# Modeling early stage atherosclerosis in a primary human vascular microphysiological system

Xu Zhang[1], Muath Bishawi [1,2], Ge Zhang[3,4], Varun Prasad[1], Ellen Salmon[1], Jason J. Breithaupt[1,5], Qiao Zhang[1] & George A. Truskey [1✉]

Novel atherosclerosis models are needed to guide clinical therapy. Here, we report an in vitro model of early atherosclerosis by fabricating and perfusing multi-layer arteriole-scale human tissue-engineered blood vessels (TEBVs) by plastic compression. TEBVs maintain mechanical strength, vasoactivity, and nitric oxide (NO) production for at least 4 weeks. Perfusion of TEBVs at a physiological shear stress with enzyme-modified low-density-lipoprotein (eLDL) with or without TNFα promotes monocyte accumulation, reduces vasoactivity, alters NO production, which leads to endothelial cell activation, monocyte accumulation, foam cell formation and expression of pro-inflammatory cytokines. Removing eLDL leads to recovery of vasoactivity, but not loss of foam cells or recovery of permeability, while pretreatment with lovastatin or the P2Y$_{11}$ inhibitor NF157 reduces monocyte accumulation and blocks foam cell formation. Perfusion with blood leads to increased monocyte adhesion. This atherosclerosis model can identify the role of drugs on specific vascular functions that cannot be assessed in vivo.

[1] Department of Biomedical Engineering, Duke University, Durham, NC 27708, USA. [2] Division of Cardiothoracic Surgery, Department of Surgery, Duke University, Durham, NC 27708, USA. [3] Department of Immunology, College of Basic Medical Science, Dalian Medical University, 116044 Dalian, China. [4] Department of Microbiology and Immunology, School of Medicine, University of North Carolina at Chapel Hill, Chapel Hill, NC 27599, USA. [5] University of Miami Miller School of Medicine, Miami, FL 33163, USA. ✉email: gtruskey@duke.edu

The pathogenesis of atherosclerosis involves the accumulation of cholesterol containing low-density lipoprotein (LDL) in the arterial wall interacting with risk factors (e.g. smoking, hypertension). Subsequent free radical generation leads to lesion formation[1,2]. The oxidative environment activates vascular endothelial cells (ECs) as evidenced by increased expression of endothelial adhesion molecules and reduced vasodilation in response to increased blood flow[3]. Plasma LDL undergoes a series of modifications and accumulates in the intima[4] (Fig. 1a). Eventually monocytes enter and differentiate into macrophages, joined by proliferation of resident macrophages[4,5]. This is a critical step in the process, as this pool of macrophages accumulate cholesterol from modified LDL in an unregulated fashion by scavenger receptors, forming foam cells[2,6]. Medial smooth muscle cells (SMCs) migrate and proliferate in the intima and also form foam cells[1,2].

Inflammation plays an important role in the development and clinical presentation of atherosclerosis[7,8], and significantly higher rates of atherosclerosis occur in individuals with autoimmune disease, such as rheumatoid arthritis (RA)[9–11]. The recent CANTOS trial demonstrated that cardiac events and death due to atherosclerosis are reduced by an antibody to IL-1β, which is used to treat RA[12] and the COLCOT study showed that the inexpensive anti-inflammatory compound colchicine leads to reduced inflammation and fewer adverse cardiovascular events[13]. These studies emphasize the potential of anti-inflammatory treatments to reduce clinical complications of atherosclerosis, but other anti-inflammatory treatments have not been effective[14]. Thus, there is a need to establish those immune pathways that can reduce clinical symptoms and aid regression of the disease.

Mechanisms of the initiation and progression of atherosclerosis have largely been studied using animal models[15,16]. These animal models however suffer from a number of limitations including important differences in disease progression than that seen in humans, lack of external validation of the results to humans, expense and long duration to assess disease progression, and the difficulty of isolating the many factors involved in such a complex disease[17,18]. Human microphysiological systems (MPS) have been introduced to address many of these pitfalls and to examine the effect of genetic variants in disease development[19,20].

There are three types of vascular MPS: (1) microvascular systems that focus on self-organization to form vascular networks in ECM (extra-cellular matrix) and stromal cells for vascular development and related disease studies[21–25]; (2) microfluidic vascular chips that mimic the vascular lumen through culture of the ECs in well-designed flow channels, to examine interaction of ECs with other cells and thrombosis[21,26–28]; and (3) tissue-engineered blood vessels (TEBVs) consisting of intimal and medial layers to study drug responses and model arterial diseases[29–31]. While some of the other vascular models included endothelial cells and medial cells, only the TEBV examined endothelial cell-medial cell interactions on vascular function in vitro[29–31].

In this work, we extend our human TEBV model and use it to simulate many key features in early atherosclerosis. We create a chip to fabricate and perfuse four TEBVs simultaneously and develop a three-layer model of ECs, SMCs, and fibroblasts. Endothelial dysfunction is an early step in atherosclerosis and plays a central role in many vascular-related disorders[32–34]. Similar to natural blood vessels, functional properties of tissue-engineered vessels are highly dependent on the endothelial layer[35,36]. Therefore, endothelization of TEBVs is important for the overall utility of this model, and examination of the impact of their functional properties[29,31,37,38].

To create a model of early stage atherosclerosis and examine the effect of drug treatments, we exposed human TEBVs to enzyme-modified low-density lipoprotein (eLDL) (Fig. 1a). The TEBVs are then tested in the presence of eLDL, the cytokine tumor necrosis factor α (TNFα), and the addition of circulating monocytes in cell media or human whole blood with notable changes in vasoactivity, permeability, endothelial activation, as well as monocyte accumulation and foam cell formation in the vessel wall, all important hallmarks of early atherosclerosis. Many of these short-term pro-inflammatory events are reversible in vitro and these early atherosclerotic events can be reduced with drug treatments.

## Results

**Fabrication and characterization of TEBVs.** Four TEBVs with 2 layers or 3 layers were fabricated simultaneously in situ in one chip using different parts of the mold (Fig. S1). To enhance the overall concentration of collagen and improve the mechanical properties of TEBVs, application of plastic compression removed over 90% of the water (Fig. S2). The final concentration of collagen in the TEBVs was 70 mg/ml, which is similar to the value in vessels in vivo[39]. Final TEBV dimensions were >10 mm in length and 1074 ± 99 μm (n = 4) in outer diameter. For the EC/fibroblast 2-layer TEBV, the lumen was circular with an internal diameter of 647.5 ± 45.7 μm (n = 4). The vessel wall had a mean thickness of 210 μm (Fig. 1c). Cross-sectional images demonstrate a well-formed endothelial cell layer (evident by CD31 staining) defining the vessel lumen, with hNDFs ±hSMCs distributed uniformly in vessel wall (Fig. 1d). *En face* (Fig. 1e) and 3D reconstructions of the TEBV (Fig. 1f) demonstrate good endothelial coverage. Furthermore, the collagen TEBV does not permit fluid extravasation as shown by perfusion of a dye through the lumen (Supplementary Movie 1).

While primary human SMCs are most relevant to accurately mimic human blood vessels, they suffer from several shortcomings including limited proliferative capacity and donor availability, and significant donor to donor variation[40]. Often donors are older and have cardiovascular disease resulting in SMCs with reduced contractility[41]. TEBVs fabricated with hNDFs are significantly more contractile than those fabricated from MSCs[31] and still express the key proteins calponin and αSMA. In 3-layer TEBVs, the layers of EC, SMC, and fibroblast were clearly distinguished (Fig. 1g), and the total medial thickness was similar to the value for the two-layer model and the inner hSMC layer (75.5 ± 10.6 μm (mean ± S.D.)) was about half the thickness of the outer hNDF layer, even though the same number of hSMCs and hNDFs were initially seeded into the collagen solution. In the 3-layer TEBVs, both hNDFs and hSMCs expressed smooth muscle-related markers αSMA and MHC11 both in cross-sections (Fig. S3a, b) and longitudinal sections (Fig. S3c).

**TEBV properties and function after perfusion.** Perfusion began within 24 h of fabrication at a flow rate of 0.5 ml/min per TEBV (0.4 Pa shear stress). The burst pressure after dehydration, 1.036 ± 0.006 bar, increased to 1.61 ± 0.03 bar after one week of perfusion and was relatively constant for the next 3 weeks (Fig. 2a). The ultimate tensile stress of the TEBVs was as high as 12–15 bar and was stable for at least 4 weeks (Fig. 2b). After one week of perfusion, TEBVs exhibited phenylephrine-induced vasoconstriction and acetylcholine-induced vasodilation (Fig. 2c), which were maintained over 4 weeks of perfusion. After 48 h perfusion nitric oxide (NO) production of nitrate and nitrite was about 20 μM in the perfusion media (Fig. 2d) and was maintained over four weeks. Endothelial release of NO further demonstrates maturation of TEBVs after 1 week of perfusion. Furthermore, the TEBV diameter is stable over time with changes in the diameter of less than ±5% with no consistent trend (Fig. 2e).

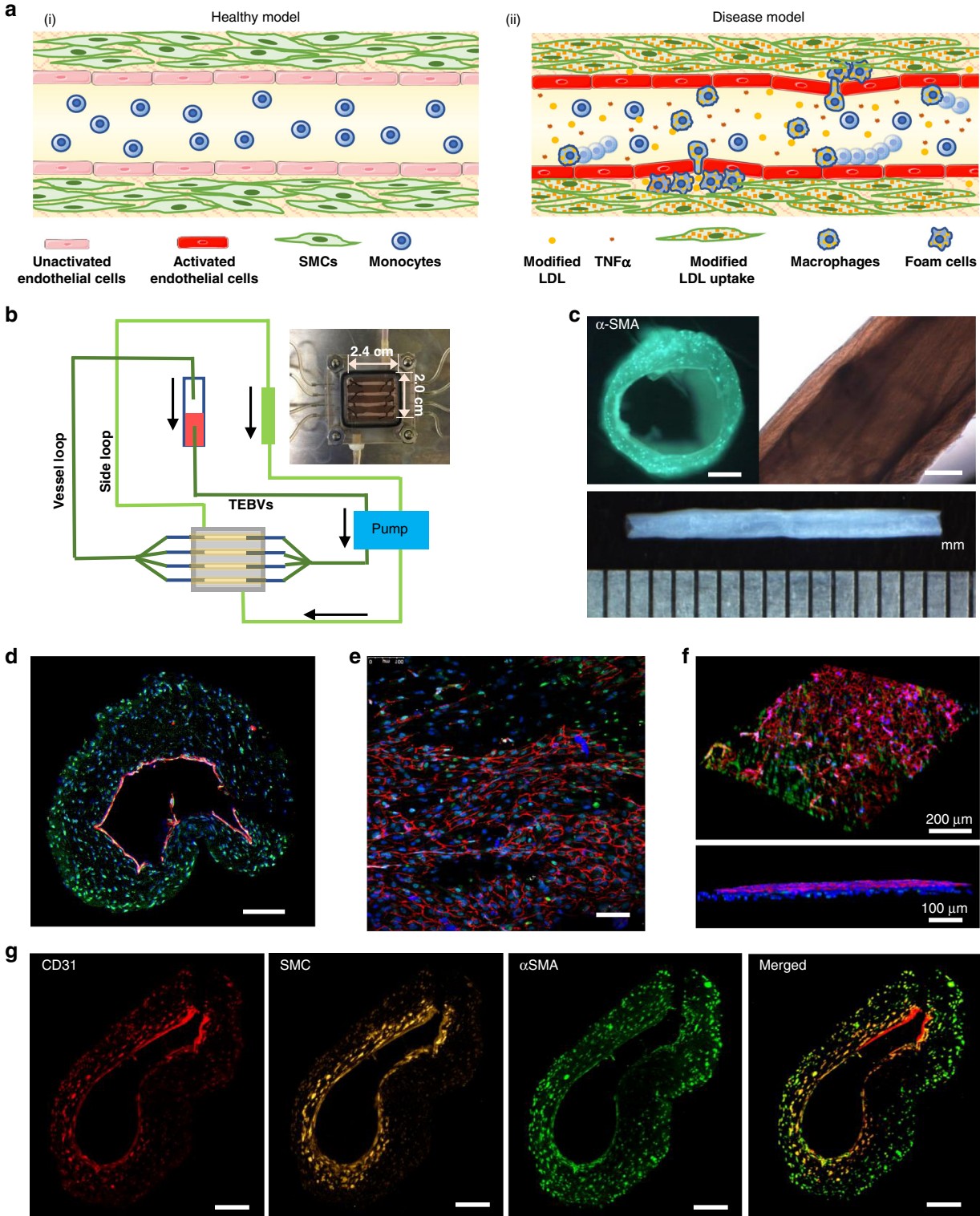

**Fig. 1 Fabrication and characterization of tissue-engineered blood vessels (TEBVs.). a** Under healthy conditions (i), monocytes do not adhere to the endothelium. Under disease conditions (ii), modified forms of LDL (low-density lipoprotein), or cytokines such as TNFα, activate the endothelium to increase expression of leukocyte adhesion molecules and release cytokines that enable monocytes to adhere to the endothelium and transmigrate into the media. Modified LDL in the media is taken up by monocytes/macrophages and hSMCs (human smooth muscle cells) becoming foam cells. **b** Schematic diagram of the perfusion system. **c** Bright-field views of TEBV. Scale bar = 200 μm. **d** Cross-sectional view of 2-layer TEBV showing an intact CD31 positive endothelium; Scale bar = 200 μm. **e** En face view of lumen showing CD31 positive endothelium overlying αSMA positive hNDF; Scale bar = 100 μm. **f** 3D view of opened TEBV (**d**–**f**: CD31-red, αSMA-green, DAPI-blue). **g** Cross-sectional view of 3-layer TEBV: endothelial layer (CD31-red), smooth muscle cell layer (cell tracker-yellow), fibroblast layer (αSMA-green); Scale bar = 200 μm.

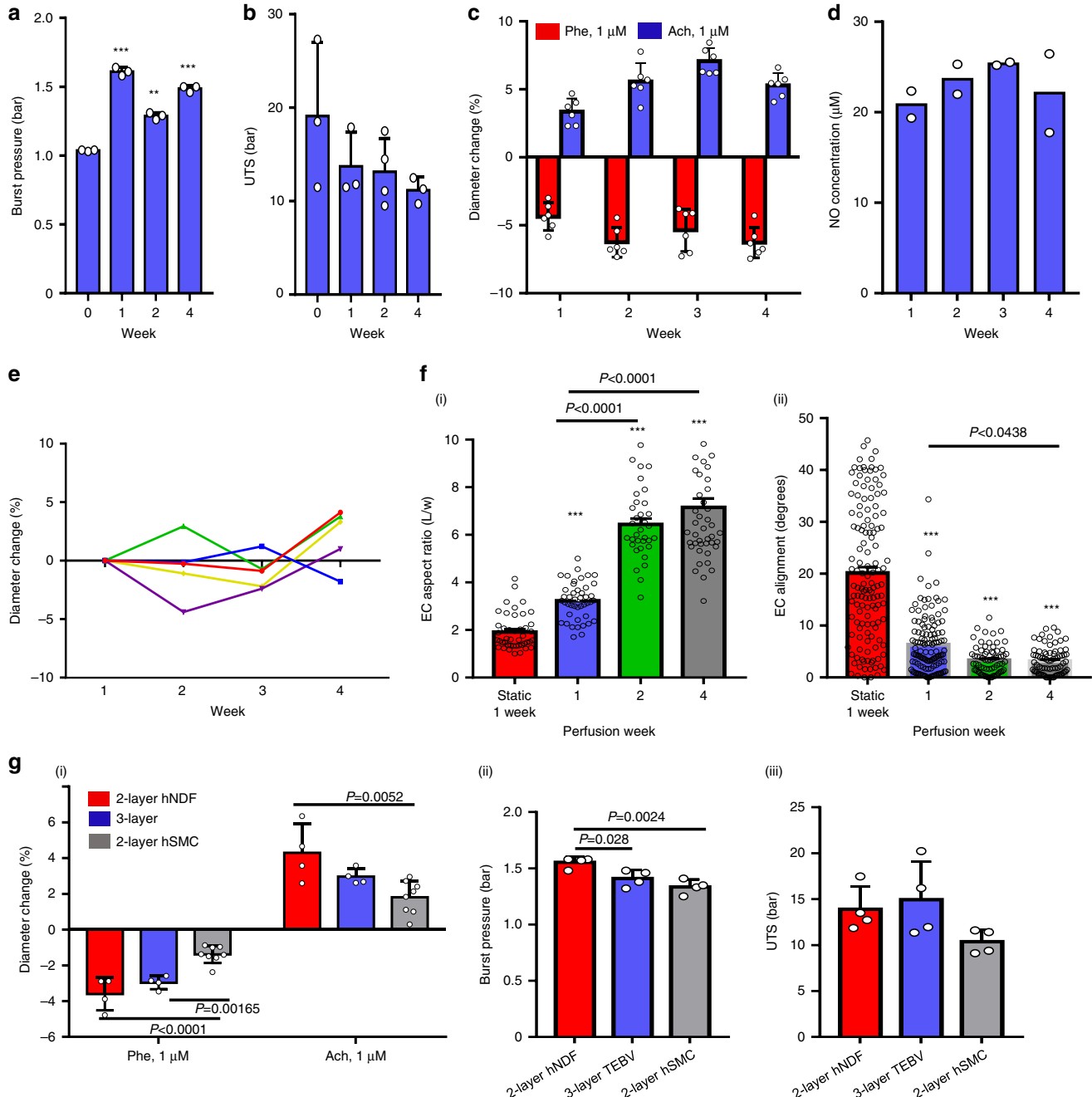

**Fig. 2 TEBV functional properties and stability during 4 weeks of perfusion.** Burst pressure (**a**) and ultimate tensile stress (UTS) (**b**) of endothelialized TEBVs perfused for 1–4 weeks (mean ± S.D., $n = 3$ TEBVs, $**P = 0.00031$, $***P < 0.0001$ compared to week 0 without perfusion by one-way ANOVA). **c** Vasoactivity of endothelialized TEBVs perfused for 1–4 weeks (mean ± S.D., $n = 6$ TEBVs), Phe: phenylephrine, Ach: acetylcholine. **d** NO production (total nitrate and nitrite concentration in media) of endothelialized TEBVs perfused for 1–4 weeks (mean ± S.D., n = 2 experiments with 4 TEBV per experiment). **e** The outer diameter is stable over 4 weeks perfusion, without treatment. Each point represents a value from a single TEBV at each time point. **f** ECs (endothelial cells) elongate (i) and align (ii) under perfusion (mean ± S.E., $n = 44,44,34,45$ cells for elongation and $n = 134,126,70,77$ cells for alignment from 2 TEBVs, $***P < 0.0001$ compared to static culture by one-way ANOVA and Tukey post hoc test). **g** Vasoactivity (i), burst pressure (ii) and UTS (iii) of 2- and 3-layer TEBVs after 1-week perfusion (mean ± S.D., $n = 8$ for 2-layer-hSMC in (i) and $n = 4$ for others, $P$ values determined by one-way ANOVA and Tukey post hoc test) (hNDF: human neonatal dermal fibroblasts; hSMC: human smooth muscle cells, Phe: phenylephrine; Ach: acetylcholine).

Compared to ECs in static culture TEBV, ECs on the luminal surface of TEBVs exposed to a shear stress of 0.4 Pa for 2 weeks perfusion were elongated and aligned with flow (Fig. 2f). ECs formed contacts with each after 1 and 2 weeks of perfusion. By 4 weeks, coverage was still very good, as noted by vWF staining, but CD31 contacts between ECs were discontinuous (Fig. S4).

To characterize the relative contractile ability of human smooth muscle cells (hSMCs) and human neonatal dermal fibroblasts (hNDFs) in TEBVs, we tested the 2-layer TEBVs with hSMCs or hNDFs and 3-layer TEBVs with both hSMCs and hNDFs. Both the 2-layer EC/hNDFs and the 3-layer (EC/hSMC/hNDF) TEBVs exhibited good vasoactivity and mechanical strength, but the 2-layer EC/hSMCs TEBVs exhibited reduced

vasoactivity and mechanical strength as the other two types of TEBVs (Fig. 2g). In all subsequent experiments, we only compared the difference of vasoactivity between 2-layer (EC/hNDFs) and 3-layer TEBVs. Other studies were all completed based on 2-layer TEBVs with hNDFs.

**Response of individual cell types to eLDL.** LDL can be modified by oxidation[42], acetylation[43], and enzymatic modification[44]. These modified forms of LDL activate ECs, enter macrophages by scavenger receptors, and induce the formation of foam cells. We used enzyme-modified LDL (eLDL), because eLDL is more effective than oxidized LDL in producing foam cells[44], causes upregulation of ICAM-1 and E-selectin in ECs[45], and readily induces foam cell formation in SMCs[46] and macrophages[44]. The eLDL production process increased LDL particle size from $20.5 \pm 2.0$ nm ($n = 2$) to $67.0 \pm 6.5$ nm ($n = 9$) which is similar to the increase reported by others[44]. To validate that eLDL modification affected the individual cells present in our early atherosclerosis TEBV model, exposure of ECs to 100 μg/ml eLDL for 24 h induced expression of VCAM-1, ICAM-1, and E-Selectin (Fig. S5b). In contrast, treatment of ECs with 100 μg/ml LDL did not produce an increase in ICAM-1 (Fig. S5b). eLDL concentrations as low as 10 μg/ml produced foam cells in macrophages after a 24 h exposure, whereas higher eLDL concentrations (50 μg/ml) were needed to produce foam cell formation in hNDFs (Fig. S5c), similar to SMCs[46]. Furthermore, exposure to 100 μg/ml eLDL for more than 48 h induced EC detachment but 50 μg/ml eLDL did not affect ECs after 96 h exposure. Based on these results, we used 50 μg/ml eLDL to treat the TEBVs.

**eLDL exposure reduces TEBV vasoactivity.** To model the early stages of atherosclerosis in the TEBV system and identify the contribution of a pro-inflammatory cytokine in early atherosclerosis development, TEBVs were exposed to a shear stress of 0.4 Pa for one week and then perfused with media containing 50 μg/ml eLDL with or without 50 U/ml TNFα (Fig. 3a). With those treatments, no changes are observed for the burst pressure and UTS of TEBVs in the different groups (Fig. 3b). Relative to untreated controls (Fig. 3c-i), vasoconstriction and vasodilation of the TEBVs was significantly reduced after 96 h of perfusion with eLDL (Fig. 3c-ii). Likewise, after 8 h of TNFα exposure, similar reductions in vasoactivity occurred (Fig. 3c-iii). The combination of both TNFα and eLDL did not have an additive effect on vasoactivity (Fig. 3c-iv). To test if these effects are reversible, after 8 days of recovery following removal of eLDL or TNFα in the perfusion media, 70% of baseline vasoactive function was recovered, a possible indication of significant recovery from eLDL specific vasoactivity changes in the early stages of atherosclerosis by correction of eLDL levels alone (Fig. 3c-ii to c-iv). The 3-layer TEBVs responded similarly to 2-layer TEBVs in control conditions and when exposed to eLDL or TNFα (Fig. S6).

Without eLDL or TNFα treatment, the TEBVs produced stable levels of NO over 12 days (Fig. 3d-i). eLDL and/or TNFα treatment led to an increase in NO production. TNFα treatment alone produced a rapid increase in NO levels, while eLDL treatment alone led to an increase in NO levels by 96 h treatment (Figs. 3d-i to d-iii). In the recovery period, both treatment arms demonstrated a drop in NO in the early recovery period, which was followed by slow re-establishment of baseline levels over time (Fig. 3d-ii and d-iii). No additive effect of co-treatment of eLDL and TNFα is observed in NO production during the early treatment or the recovery period was observed (Fig. 3d-iv).

**eLDL exposure increases TEBV permeability to macromolecules.** To test the permeability of the vessels with and without eLDL

treatment and following recovery after removal of eLDL, 20 μg/ml of FITC labeled dextran 500 kDa (Stokes–Einstein radius = 15.9 nm)[47] or 10 μg/ml of FITC labeled goat IgG (Stokes–Einstein radius = 5.6 nm)[48] was added to the media perfused through the TEBVs, and fluorescence images were captured every 5 min. Dextran and IgG test the sensitivity of the permeability to molecular weight and size. Dextran diffused into the vessel wall rapidly in the TEBVs without cells (Fig. S7ai), but in the perfused TEBVs with endothelium, dextran was confined to the region near the interface between the TEBV and the lumen (Fig. S7aii and S7aiii). The permeability of static culture TEBVs was lower than no-cell TEBVs but higher than perfused TEBVs, which demonstrated that perfusion increases the EC junction integrity (Fig. 4b). The permeability of TEBVs to dextran with and without the endothelial layer is $5.9 \pm 1.4 \times 10^{-7}$ cm/s and $2.8 \pm 0.1 \times 10^{-6}$ cm/s at 10 min and $6.3 \pm 2.3 \times 10^{-7}$ cm/s and $3.3 \pm 0.4 \times 10^{-6}$ cm/s at 20 min (Fig. 4a). The permeability of TEBVs significantly dropped with the endothelial layer present, which served as additional evidence for the presence of functional endothelial layer. After eLDL exposure for 96 h, the permeability of endothelial TEBVs increased to $2.3 \pm 1.7 \times 10^{-6}$ cm/s at 10 min and to $3.0 \pm 1.8 \times 10^{-6}$ cm/s at 20 min, which is about 5 times higher compared to the control group (Fig. 4a). After 8 days of recovery post eLDL exposure, the permeability of TEBVs was still $2.1 \pm 1.3 \times 10^{-6}$ at 10 min and $1.7 \pm 1.2 \times 10^{-6}$ at 20 min, which revealed the barrier function of TEBVs did not recover in 8 days (Fig. 4a). The IgG permeability of TEBVs was similar to dextran 500 kDa in all the groups. Thus, the barrier function of TEBVs was stable to different large molecules (radius 5–15 nm) and was impaired by eLDL.

**Inflammation in TEBVs similar to early stages of atherosclerosis inflammation.** Early inflammatory changes are a crucial factor in atherosclerosis development[7,8]. In the absence of eLDL or TNFα, VCAM-1, ICAM-1, and E-selectin expression were absent or present at very low levels (Fig. 5a). Activation of endothelial cells after eLDL and TNFα addition to the perfusion media was characterized by increased expression of VCAM-1, ICAM-1, and E-selectin (Fig. 5a) at flow rates of 0.125 ml/min (0.1 Pa) per TEBV and 0.5 ml/min (0.4 Pa) per TEBV. Treatment with eLDL led to slight increases in the expression of ICAM-1 and E-selectin at both flow rates, while TNFα treatment led to upregulation of VCAM-1 and E-selectin at 0.4 Pa (Fig. 5b). When the perfusion media contained 50 μg/ml eLDL for 96 h and 50 U/ml TNFα during the last 8 h of eLDL perfusion, all three adhesion molecules were elevated relative to controls. After 8 days of recovery (0.1 Pa), the activation of endothelial cells subsided in all the groups (Fig. 5a, b), further suggesting early stages of atherosclerosis may be partly reversible. Furthermore, eLDL and TNFα treatment led to increased expression of TNFα and IL-1β in the TEBVs (Fig. 5c) with a synergistic response to the combination of eLDL and TNFα. Expression of TNFα and IL-1β declined after 8 days of recovery but remained 2–3 times higher than controls.

Next, we examined the expression level of ECM genes Col I, Col III, and Col IV in the TEBVs. Col I and Col IV expression did not change following 96 exposure to 50 μg/ml eLDL or 8 h TNFα treatment (Fig. 5d). After removal of eLDL from the perfusion media for 8 days, Col I and Col IV gene expression increased.

**Monocytes accumulation in TEBVs after eLDL exposure.** Monocyte/macrophages play an important role in the early stages of atherosclerosis[49]. After establishing that the individual cell types responded to eLDL, we examined the effect of different eLDL concentrations and exposure periods to cause substantial increases in monocyte accumulation. Adding 50 or 100 μg/ml eLDL to the perfusion media for 48 h and adding

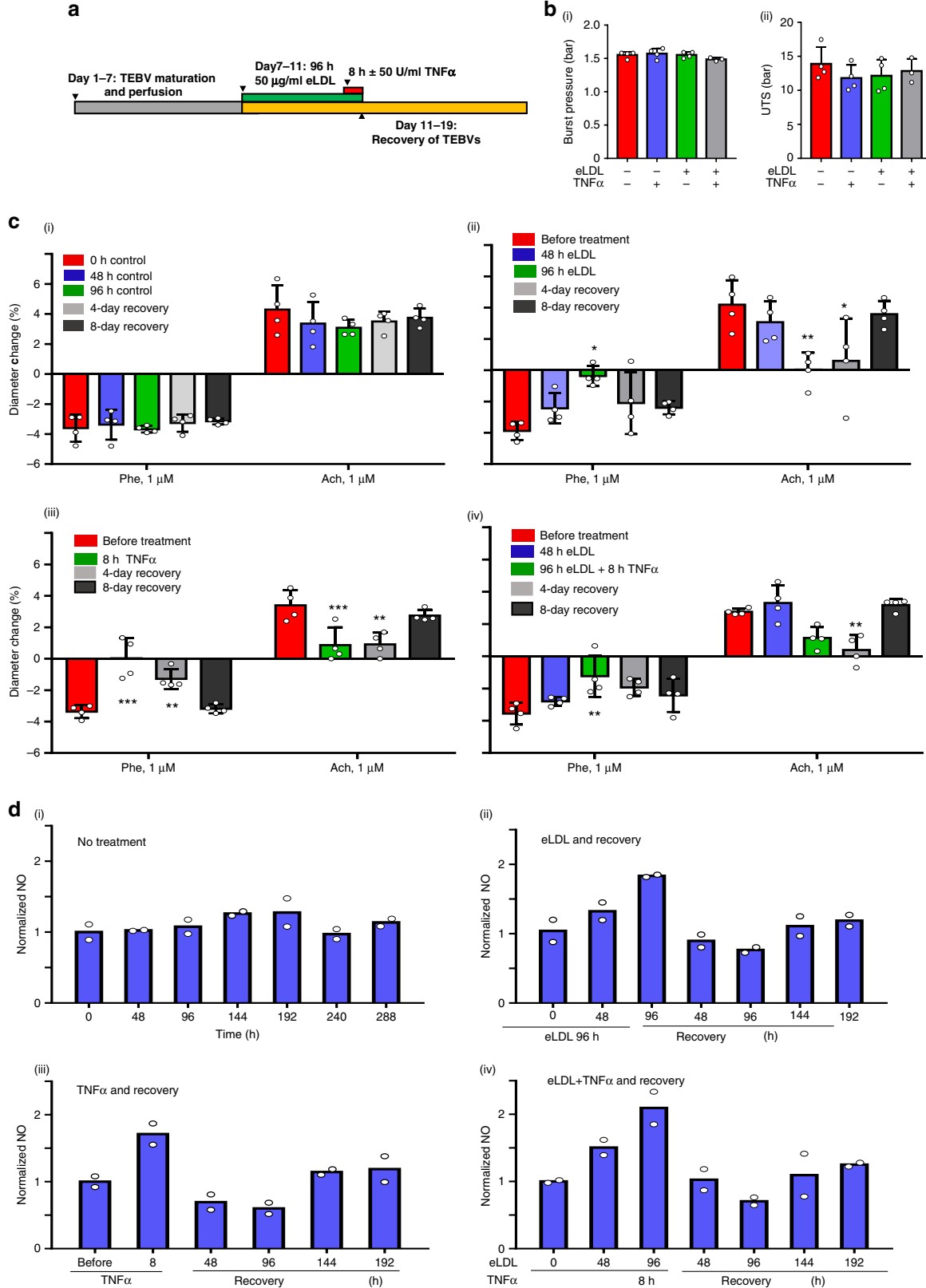

$1 \times 10^6$ U937 monocytes/ml during the last 24 h of eLDL exposure (Fig. 6a) produced a modest increase in monocyte accumulation, which increased greatly if 50 U/ml TNFα is added during the last 8 h (Fig. 6b).

We next investigated monocyte accumulation in TEBVs following longer eLDL and monocyte exposure periods. After

one week of maturation, TEBVs are treated with 50 μg/ml eLDL for times up to 96 h with or without 100 U/ml TNFα for the last 8 h of perfusion with eLDL (Fig. 6c). After 72 h of perfusion at 0.125 ml/min per TEBVs (0.1 Pa), labeled monocytes accumulated on the luminal endothelial cells. Few monocytes adhered in the absence of eLDL or TNFα (Fig. 6d, e). The number of

**Fig. 3 Dysfunction and recovery of TEBVs by eLDL and/or TNFα exposure. a** Timeline of TEBV treatment. **b** Burst pressure (i) and UTS (ii) of TEBVs after 96 h eLDL ± 8 h TNFα exposure (mean ± S.D., $n = 3$ TEBVs for +/+ and $n = 4$ TEBVs for others). **c** (i) Vasoactivity for perfusion without treatment. (ii) Vasoactivity for perfusion with 50 µg/ml eLDL followed by 8 days of recovery without eLDL in perfusion media (mean ± S.D., $n = 4$ TEBVs, Phe: *$P = 0.0335$; Ach: *$P = 0.028053$; **$P = 0.006529$ compared to before treatment by two-way ANOVA and Tukey post hoc test). (iii) Vasoactivity for perfusion for 8 h with 50 U/ml TNFα exposure followed by 8 days of recovery without TNFα (mean ± S.D., $n = 4$ TEBVs, Phe: **$P = 0.00246$, ***$P < 0.0001$; Ach: **$P = 0.0049$, ***$P = 0.0041$; Ach **$P = 0.0049$, compared to before treatment by two-way ANOVA and Tukey post hoc test). (iv) Vasoactivity for perfusion with 50 µg/ml eLDL for 96 h and 50 U/ml TNFα for 8 h and 8 days of recovery (mean ± S.D., $n = 4$ TEBVs, Phe:**$P = 0.0081$; Ach **$P = 0.0065$ compared to before treatment by two-way ANOVA and Tukey post hoc test). Phe: phenylephrine, Ach: acetylcholine (**c**). **d** Normalized NO production (total nitrate and nitrite concentration in media) of TEBVs with no treatment (i), 96 h of eLDL (ii), 8 h of TNFα (iii), 96 h eLDL + 8 h TNFα (iv) exposure and 8 days of recovery ($n = 2$ from 8 TEBVs).

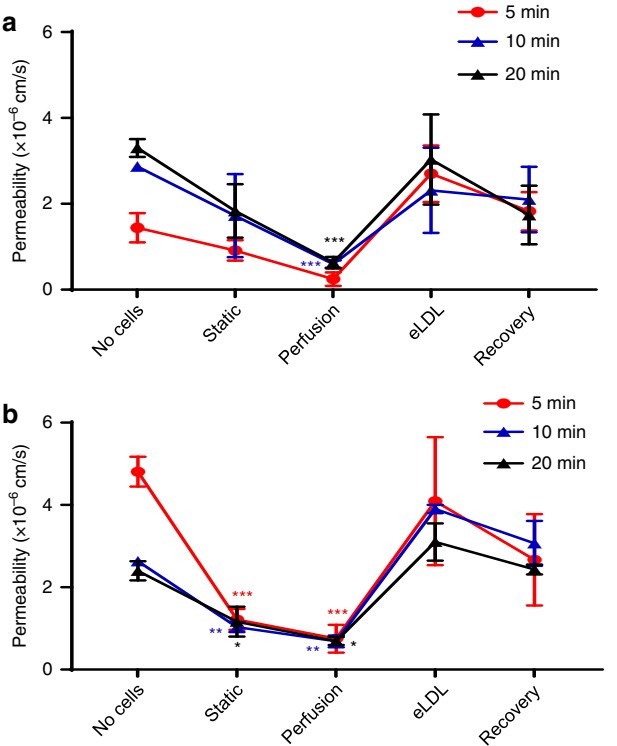

**Fig. 4 Perfusion reduces TEBV EC permeability.** Permeability of TEBVs to 20 µg/ml of FITC-dextran (500 kDa) (**a**) and to 10 µg/ml of FITC-IgG (**b**). The groups are collagen tubing without cells and without perfusion (No cells), TEBVs with cells under static culture of 1 week (Static), TEBVs with cells for 1-week perfusion perfused at 0.5 ml/min per TEBV (0.4 Pa) (perfusion), TEBVs under 1-week perfusion followed by 96 h perfusion with media containing 50 µg/ml eLDL (eLDL), TEBVs following 96 h exposure to 50 µg/ml eLDL followed by 8 days without eLDL (Recovery) (mean ± S.D., $n = 3$ TEBVs, (**a**) perfusion: ***$P = 0.00066$ (10 min), ***$P = 0.000105$ (20 min), (**b**) static ***$P = < 0.0001$ (5 min), **$P = 0.00785$ (10 min), *$P = 0.0358$ (20 min), Perfusion: ***$P < 0.0001$ (5 min), **$P = 0.00161$ (10 min), *$P = 0.0488$ (20 min), compared to no-cell group by a two-way ANOVA and Tukey post hoc test).

monocytes adhered on eLDL treated TEBVs was 50–100% higher than for control vessels (Fig. 6f). Co-treatment of TNFα and eLDL substantially enhanced the accumulation of monocytes up to 3–4 times the control levels (Fig. 6f). At a higher flow rate of 0.5 ml/min per TEBV (0.4 Pa) monocyte adhesion was reduced in control TEBVs but had little effect on TEBVs exposed to either eLDL or TNFα. To the eLDL + /TNF + treatment, the accumulation of monocytes under high shear stress declined about 25% compared to the low shear stress case, but was still higher than the groups with eLDL or TNFα treatment.

Migration of monocytes into the vessel wall was also observed (Fig. 6g). After 72 h prefusion, the migration distance of monocytes is about 30–50 µm within the vessel wall (Fig. 5e), a process similar to what is seen in vivo[1].

Next, we perfused whole blood with 25–30% CPD through the TEBV lumen (Fig. S8). Although the blood contained CPD to prevent clotting, perfusion of whole blood for more than 40 h did lead to obstruction of the TEBVs and the flow rate declined significantly. We modified the protocol to perfuse the TEBVs for 7 days with the perfusion media as before (Fig. 6h). Next, the media was replaced with whole blood with or without 50 µg/ml eLDL which was perfused at 0.5 ml/min (0.4 Pa) for 24 h, after which $10^6$/ml cell tracker labeled monocytes were added and the blood perfused for another 40 h at 0.125 ml/min (0.1 Pa). A shear stress of 0.1 Pa approximates the mean value at vessel branches where atherosclerosis originates[50]. A control without blood was performed simultaneously.

Compared to perfusion with culture media, perfusion with whole blood increased monocyte accumulation in the TEBVs ~3.4X without eLDL treatment (Fig. 6i). When TEBVs were exposed to whole blood containing eLDL, monocyte adhesion increased another 50%.

**Foam cell formation in TEBVs.** To characterize foam cell formation, TEBVs treated as in Fig. 6c were stained with Oil Red-O. Oil Red O + monocytes were adherent to the vessel lumen (Fig. 6j, k) as well as within the TEBVs. The hNDFs in TEBVs accumulated eLDL after 96 h exposure and many Oil Red O + regions are observed in the cells (Fig. 6l and Fig. S9). Oil Red O + regions in the endothelial layer were rare after 96 h eLDL exposure, which revealed that ECs deliver but do not accumulate eLDL (Fig. S9iii). This result is consistent with in vivo studies with mice[51].

However, U937 monocytes did not express CD80, an important marker of M1 phase macrophages[52], consistent with other reports[53]. Next, we examined primary monocyte polarization into macrophages in both 2D and the TEBV system. In 2D, primary monocytes were tested with or without pretreatment of Granulocyte-Macrophage Colony Stimulating Factor (GM-CSF). GM-CSF promotes monocyte polarization to pro-inflammatory macrophages and dendritic cells[54]. Without GM-CSF pretreatment, levels of CD80 were low (CD80 expression was 15–22% after eLDL and/or α treatment). Further, eLDL and/or TNFα treatment did not increase the number of CD80 + cells. Thus, we pretreated the primary monocytes with 20 ng/ml GM-CSF for three days, which increased the baseline of CD80 expression from <10% to >40% and increased the primary monocytes survival ratio in vitro. GM-CSF treated monocytes were then exposed to 50 µg/ml eLDL and/or 50U/ml TNFα for 72 h. eLDL and TNFα treatments increased expression of CD206, a common marker for polarization into macrophages[52]. TNFα alone, eLDL alone,

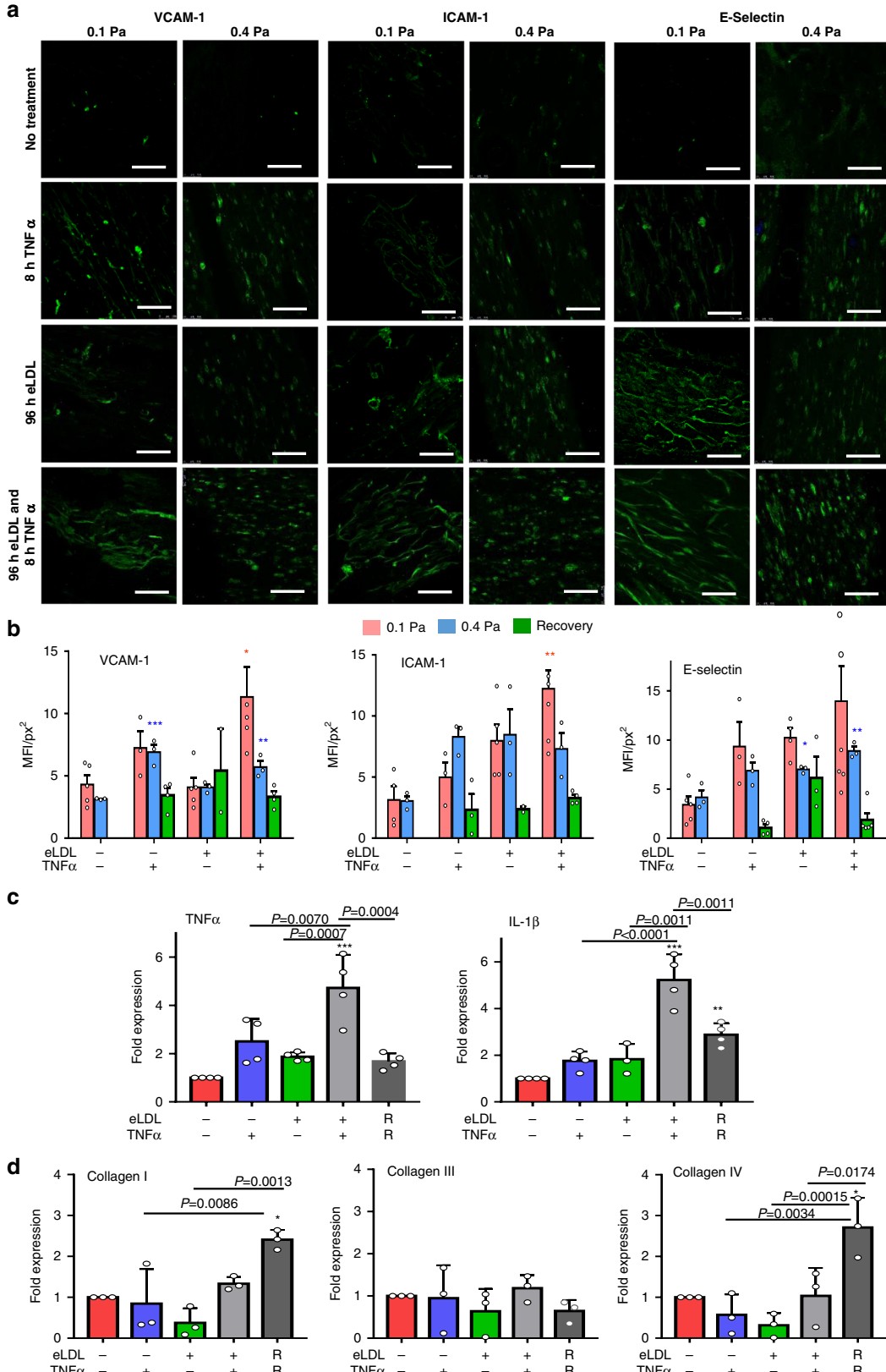

and both combined had differential effects on CD80 expression (Fig. 7a, b). Expression of CD36, an important scavenger receptor is greatly increased in the treatment groups (Fig. 7a, b). This receptor present on monocytes and macrophages has an important function in lipid uptake and foam cell formation[55,56].

Next, TEBVs were perfused with media, TNFα, or eLDL with or without TNFα as shown in Fig. 6c. Following 72 h perfusion at 0.125 ml/min per TEBV (0.1 Pa) with GM-CSF-treated primary monocytes labeled with cell tracker red, monocytes that accumulated in the TEBVs were characterized through

**Fig. 5 Inflammatory response of TEBVs following exposure to eLDL or TNFα. a** Fluorescence images of VCAM-1, ICAM-1 or E-selectin expression on TEBV ECs after flow of 0.125 ml/min per vessel (0.1 Pa) and 0.5 ml/min per vessel (0.4 Pa) for 8 days followed by exposure to media alone for 96 h, media with 50 µg/ml eLDL for 96 h, media alone for 96 h followed by media and 50 U/ml TNFα for the last 8 h, or media with 50 µg/ml eLDL for 96 h and 50 U/ml TNFα for the last 8 h (scale bar: 200 µm). **b** Quantification of mean fluorescence intensity (MFI) of the images in 5a and following the recovery for 8 day at 0.5 ml/min (mean ± S.E., n = 5,3,3,3,4,5,3,2,5,3,4 TEBVs for VCAM-1 accordingly, n = 4,3,3,3,3,5,3,2,7,3,4 TEBVs for ICAM-1 accordingly, n = 5,3,3,3,4,4,3,3,7,3,5 TEBVs for E-Selectin accordingly, VCAM-1 0.1 Pa *P = 0.0195,0.4 Pa, 0.4 Pa **P = 0.0094, ***P = 0.0008; ICAM-1 0.1 Pa **P = 0.0021, 0.4 Pa; E-selectin 0.4 Pa *P = 0.0448, *P = 0.0027 relative to the respective control by one-way ANOVA and Tukey post hoc test at each shear stress). **c** RT-PCR of TNFα and IL-1β (mean ± S.D., n = 3 TEBVs for +/−in IL-1β n = 4 TEBVs for others,**P = 0.0069, ***P < 0.0001, compared to control by two-way ANOVA) and (**d**) RT-PCR collagen I, III, and IV expression (mean ± S.D., n = 3 TEBVs, Col I: *P = 0.0169, Col IV: *P = 0.0152 compared to control by one-way ANOVA and Tukey post hoc test) after treatment with eLDL for 96 h and/or TNFα for 8 h, and 8 days of recovery (R) after treatment with LDL and/or TNFα at a flow rate of 0.5 ml/min.

CD80 staining. All CD80 positive cells (green in Fig. 7c) are positive for cell tracker red. CD80 expression of macrophages accumulating in the TEBVs was similar to experiments with isolated monocytes (Fig. 7b) in which eLDL leads to significant increase in the CD80 expression, with a smaller increase for TNFα only treatment (Fig. 7c, d). An additional additive effect in CD80 expression is seen with both eLDL and TNFα co-treatment.

**Inhibition of effect of eLDL on TEBV function.** Lovastatin is a widely used hydroxymethylglutaryl coenzyme A (HMG-CoA) reductase inhibitor, that decreases mevalonate production, an important part of cholesterol synthesis[57]. Treatment of the perfusion media with Lovastatin together with eLDL blocked the reduction in vasoactivity induced by eLDL (Fig. S8ai) and eLDL-induced NO production (Fig. S8aii). Addition of Lovastatin inhibits the uptake of eLDL in hNDFs and foam cell formation (Fig. S10). Lovastatin treatment blocked the effect of TNFα exposure on TEBV vasoactivity, with almost no change in NO production (Fig. 8bi, ii). Lovastatin treatment significantly decreases monocyte accumulation in TEBVs treated with eLDL, with no effect seen for TNFα treatment (Fig. 8c). Reduced monocyte adhesion and accumulation may be related to the marked decrease in ICAM-1 and E-selectin expression when eLDL exposed vessels are treated with Lovastatin (Fig. 8d). Furthermore, Lovastatin blocked CD80 expression induced by both eLDL and TNFα (Fig. 8e), and CD36 expression induced by eLDL, or TNFα (Fig. 8f).

Next, we examined whether blocking the P2Y$_{11}$ receptor with NF157 altered vasoactivity[58], EC inflammation and monocyte accumulation. The P2Y$_{11}$ receptor is a G-protein coupled receptor for ATP and is present in humans but not mice or rats[58]. Blockade with NF157 inhibited inflammation on endothelial cells induced by oxidized LDL[59]. In control experiments, exposure to 50 µg/ml eLDL inhibited vasoconstriction after 96 h exposure and inhibited vasodilation by 48 h exposure and TEBV vasoactivity recovered after 8 days (Fig. 9a). Addition of 25 µM NF157 with 50 µg/ml eLDL, caused a delayed inhibition of vasoconstriction and vasodilation with full recovery by 6 days after removal of eLDL (Fig. 9b). When monocytes were added during the last 72 h of perfusion at 0.5 ml/min (0.4 Pa), addition of 25 µM NF 157 caused a significant reduction in monocyte accumulation (Fig. 9c) and slightly reduced expression of VCAM-1, ICAM-1 and E-selection (Fig. 9d).

**Discussion**
In this work, we established a rapid and reproducible method to generate tissue-engineered blood vessels to model early stages of atherosclerosis, including eLDL-induced atherogenesis. Plastic compression of collagen TEBVs provides burst pressure and ultimate tensile strengths similar to veins and TEBVs generated with synthetic materials[4]. After endothelization and perfusion,

the TEBVs exhibited key vessel functions, including vasoactivity, NO production, and low permeability. The permeability values measured in this study are similar to those obtained with human umbilical vein endothelial cells[21,60,61] and human dermal microvascular endothelial cells[25] forming the luminal surface of vascular networks in hydrogel scaffolds exposed to physiological shear stresses for 4–7 days. The physiologically relevant microenvironment generated in our system includes physiologically relevant shear stress and flow conditions, an important element in accurately modeling vascular health and disease states. This is especially true given the significant role that shear stress plays in endothelial response to injury and disease progression.

The TEBV system was then used to model early stages of atherosclerosis, with controlled introduction of different elements characteristic of the disease, including increased concentrations of eLDL, leading to cellular uptake and foam cell formation, monocyte activation and accumulation, an inflammatory environment leading to activation of ECs and monocytes, and early effects on vasoactivity[1,2,6]. After perfusion of TEBVs for 96 with 50 µg/ml eLDL, the ECs exhibited properties similar to those reported previously, including EC expression of E-selectin and ICAM-1[45,62] which then led to fibroblast and macrophage foam cells.

Since we used a straight segment of TEBV, flow rates used produce shear stresses that favor promotion of atherosclerosis[63]. Nonetheless, without eLDL or TNFα in the media, leukocyte adhesion molecule expression was very low with minimal monocyte adhesion. While a 48 h perfusion with 50 µg/ml eLDL caused a modest increase in monocyte accumulation, 96 h treatment with 50 µg/ml eLDL leads to significant accumulation of monocytes in TEBVs, reduced vasoactivity, elevated NO levels, and elevated endothelial permeability. These changes are associated with increased gene expression of IL-1β and TNFα in the TEBVs. The flow rates of 0.1 Pa and 0.4 Pa in vessels only had effect on the monocytes accumulation in the control group but have less effect to eLDL/TNFα treatment groups. This result may demonstrate that for healthy people, the exercise to increase blood flow rate could reduce the monocytes adherence in vivo, but with disease, monocytes can still adhere to the endothelium and accumulate. Interestingly, foam cell formation with eLDL preceded macrophage polarization, at least as judged by CD80 expression.

Addition of eLDL and TNFα together produced synergistic increases in adhesion molecule levels and gene expression of IL-1β and TNFα, as well as monocyte accumulation. A recent large-scale clinical trial demonstrated the efficacy of IL-1β blockade in decreasing cardiovascular events[12]. Furthermore, whole blood increases monocyte adherence compared to media, which resulted from the complex composition of blood. This phenomenon may explain the delayed response of ECM gene expression levels to eLDL and TNFα in Fig. 5c.

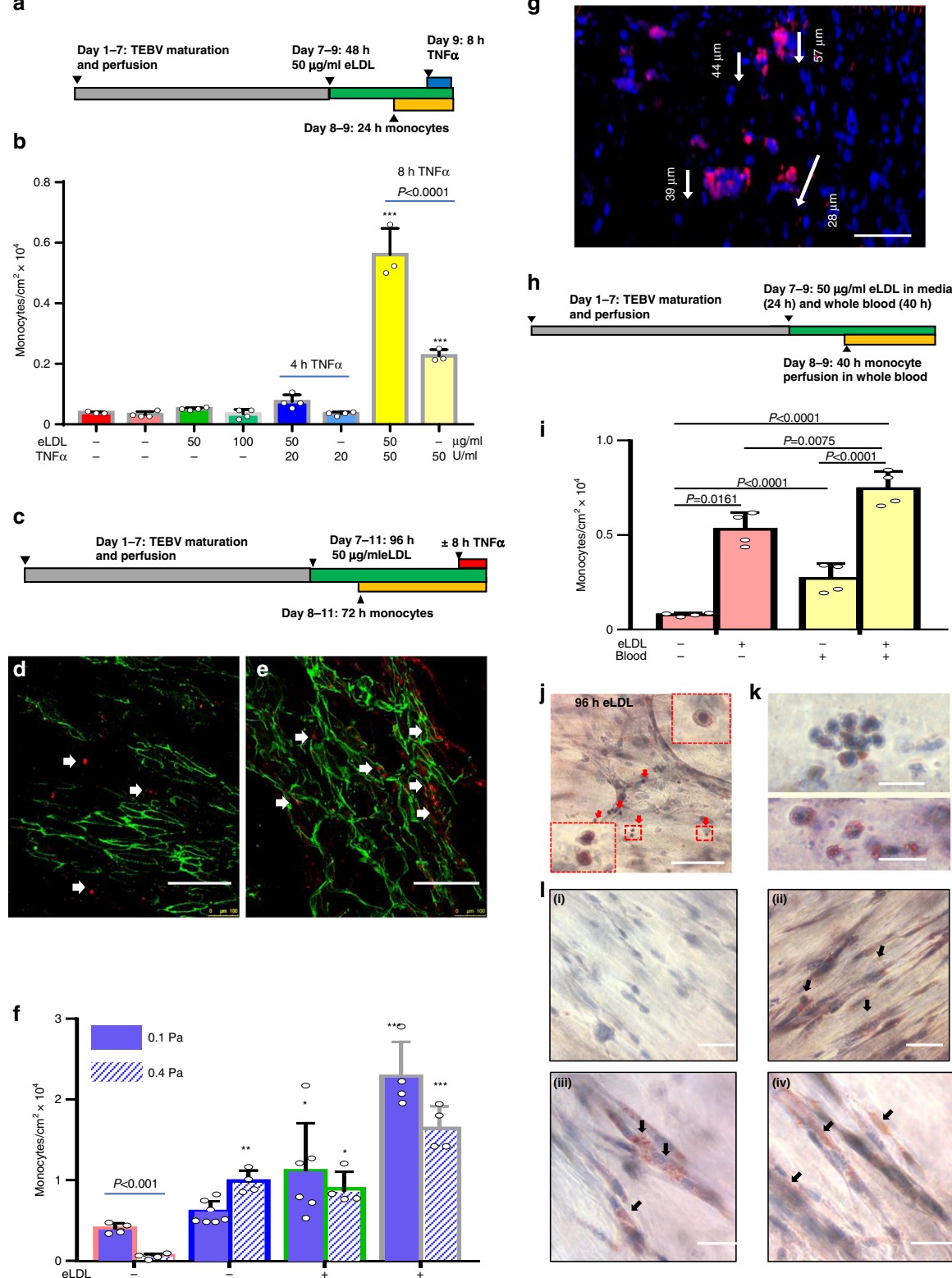

The TEBV system we developed offers several advantages. The compact chip, with replicates in a single device, enables multiple conditions to be examined using several chips run separately in an incubator. We designed the TEBVs with this dimension (OD: 1 mm, ID: 0.6 mm) because it is at the same level of a small artery and larger arteriole (mm level) and the perfusion chips with 4 TEBVs are small enough to be imaged clearly under

stereomicroscope. Further, the human TEBV microphysiological system can isolate specific effects that are difficult to assess in vivo, such separate a direct effect of statins on the vessel wall from their cholesterol-lowering effect. By keeping the diameter small, media volumes and flow rates can be maintained at small values to replicate physiological shear stresses (ml/min or less). Small volumes are ideal to measure any metabolites, secreted

**Fig. 6 Accumulation and differentiation of monocytes in the TEBV atherosclerosis disease model. a** Experimental design of 48 h TEBV treatment with eLDL. $10^6$ cell/ml of cell-tracker-red labeled U937 monocytes were added to culture media without and with 50 µg/ml eLDL and monocyte accumulation in TEBVs measured after 48 h. 50 U/ml TNFα added during last 8 h in some experiments with or without eLDL. **b** Quantification of monocytes/cm$^2$ from *en face* sections of TEBVs ($n = 3$ TEBVs for −/−, 50/50, −/50 and $n = 4$ TEBVs for others, Mean ± S.D.) ***$P < 0.0001$ relative to control by one-way ANOVA and Tukey post hoc test. **c** Experimental design of 48 h TEBV treatment with eLDL. *En face* views of monocyte (cell tracker-red) accumulation (arrow) in TEBVs without (**d**) and with 50 µg/ml eLDL (**e**) in perfusion media (scale bar 200 µm). CD31 positive endothelium shown in green. **f** Quantification of monocyte accumulation in TEBVs for different treatment conditions at flow rates of 0.125 ml/min per vessel (0.1 Pa) and 0.5 ml/min (0.4 Pa) per vessel (mean ± S.D., $n = 4,7,6,4$ TEBVs for 0.1 Pa and $n = 4$ TEBVs for 0.4 Pa; 0.1 Pa *$P = 0.0262$, ***$P < 0.0001$; 0.4 Pa *$P = 0.0146$, **$P = 0.005$, ***$P < 0.0001$ compared to control−/− by two-way ANOVA and Tukey post hoc test). **g** Migration of monocytes (cell tracker-red) in the TEBVs. Depth of monocytes into vessel wall from confocal z-sections (scale bar 100 µm). **h** Experimental design of TEBV treatment with eLDL (40 h) with and without whole blood (40 h). **i** Quantification of monocyte accumulation in TEBVs after treatment with eLDL and whole blood as in panel h (mean ± S.D., $n = 4$ TEBVs, P values obtained by two-way ANOVA and Tukey post hoc test)). **j–l**. Oil-red O positive macrophage foam cells in TEBVs (scale bar: 100 µm in (**j**) and 20 µm in (**k**). **l** Fibroblasts uptake of eLDL detected by Oil Red O staining for: control (i), 96 h of eLDL (ii), 96 h of eLDL+8 h of TNFα (iii), 8 days of recovery after eLDL treatment (iv) (scale bar: 50 µm).

molecule or cytokines produced and when evaluating drug responses. A limitation to any microphysiological system, either TEBV or microfluidic device, to simulate in vivo conditions for larger vessels is that dynamic similarity results in higher shear stresses than those occurring in human blood vessels, which may limit leukocyte adhesion and endothelial function. If, however, the focus is upon the shear stress acting on endothelium, then the Reynolds numbers and Womersley will be less than those in vivo, potentially leading to greater leukocyte adhesion and transmigration.

A novel observation of this work is that eLDL treatment of the TEBVs inhibited vasoconstriction as well as endothelial-mediated vasodilation, which was blocked by statins and partially inhibited by NF157. Evidence that modified LDL inhibits endothelial-mediated vasodilation has been indirect, based largely on in vitro studies that oxidized LDL or its components inhibit nitric oxide production[64] and the anti-inflammatory effect of statins. Prior studies that examined the effect of modified LDL upon vessel vasoactivity used isolated arterial rings and were confined to short incubation times (45–60 min). One study did report that that arterial rings treated with oxidized LDL in vitro did have attenuated vasodilation following vasoconstriction with phenylephrine[65]. Modified LDL had limited effects on vasoconstriction[66–68], although one study found that oxidized LDL enhanced vasoconstriction[65]. Differences between these studies and current results may be due to the duration of exposure or the type of LDL modification. These prior studies could not examine long-term recovery after exposure to modified LDL as we have shown.

This TEBV microphysiological system overcomes many of the previously discussed challenges of using animal models, by utilizing iPS or primary human cells where therapeutic effects can be tested directly. Furthermore, important hypotheses related to disease progression can be tested in this system, given the ability to control different aspects of the disease. For instance, it continues to be greatly debated if foam cells are derived from circulating monocytes being recruited to atheroprone sites, or if they are resident cells eventually transforming to foam cells, or a possible combination of the two[69,70]. Future experiments can provide macrophages as part of the construct or in circulation to help better understand how different methods of macrophage introduction change foam cell formation timing and consistency.

Other clinically relevant observation in our model relate to the timing of exposure, measured effect and eventual recovery from eLDL and TNFα exposure. eLDL induction of vessel inflammation was more long term compared to TNFα, with a more pronounced effect on ICAM-1 as compared to TNF primarily upregulation of VCAM-1. Importantly, after withdrawal of these factors, the vessel vasoactive response recovered. Lovastatin treatment in eLDL exposed vessels leads to a decrease in lesion

development and adhesion molecules expression. This was not seen with TNFα, where changes in VCAM-1, ICAM-1 and E-Selectin with Lovastatin treatment were not statistically significant. This possibly highlights the need for anti-inflammatory drugs that might be important to blocking the inflammatory specific component of atherosclerosis.

In summary, we used endothelialized arteriole-scale TEBVs to develop a model of early stage atherosclerosis in vitro. Through perfusion of enzyme-modified LDL, TNFα and monocytes in this system, the disease model of early stage atherosclerosis was established in vitro. Key features of this disease model included the dysfunction of vessels (vasoactivity and permeability), the inflammatory in vessel (increasing NO production and the activation of endothelial cells), the accumulation and migration of monocytes, and the formation of foam cells. Removal of eLDL caused partial regression of the disease. Furthermore, some novel results were observed, such as eLDL-induced dysfunction of vasoactivity, recovery of early stage atherosclerosis, and the effects of Lovastatin and NF157 in early stage atherosclerosis modeling. This work demonstrates the potential of this in vitro TEBV disease model

## Methods

**Cells isolation and culture**. All primary human cell isolations were performed using a protocol approved by the Duke University Institutional Review Board (IRB). Human umbilical cord blood within 48 h of collection was obtained from the Carolina Cord Blood Bank. All patient identifiers were removed prior to receiving the cord blood. Primary human endothelial colony forming cells (ECFCs) were isolated and expanded from umbilical cord blood and used at passage 3–6 using an established protocol in our lab[71]. ECFCs were maintained in Endothelial Cell Medium (Cell Application) with 1% Antibiotic-Antimycotic (Gibco).

Primary hNDF (Clonetics) were cultured in hNDF medium comprised of Dulbecco's Modified Eagle's Medium (DMEM) with 4.5 g/L glucose (Gibco) supplemented with 10% HI-FBS (Hyclone), 1 × Non-essential amino acids (NEAA, Gibco), 1% Pen/Strep, 1 × Glutamax (Gibco), 1 × sodium pyruvate (Gibco), and 0.1% β-mercaptoethanol (Gibco). Cells in passages 4–10 were used for all experiments.

Human Coronary Artery Smooth Muscle (hCASMCs) were purchased from Cell Applications, Inc. (San Diego, CA) and cultured in SMC media (Cell Applications 311–500). Cells were used to fabricate TEBVs at passage 6 or 7.

Primary human monocytes were isolated from whole blood with Magnetic Assisted Cell Sorting (MACS, CD14 + Beads, Miltenyi Biotec). The cells were more than 90% CD 14+ after isolation and were maintained in monocyte medium comprised of RPMI-1640 medium (Sigma) supplemented with 10% heat-inactivated FBS (Hyclone). To produce macrophages, 20 ng/ml GM-CSF (PeproTech) was added into the culture media for 3 days.

The human monocytic cell line U937 (Sigma) was maintained in RPMI-1640 medium (Sigma) supplemented with 10% FBS (Hyclone), 1 × GlutaMAX-1 (Sigma).

**Design and fabrication of the molds and perfusion chamber**. The molds used in this work were created using acrylic and composed of 5 parts, which form the fabrication mold and prefusion chamber (Fig. S1). The TEBV (2 layers) fabrication mold is formed from parts A, B, and C. Part A housed the mandrels about which

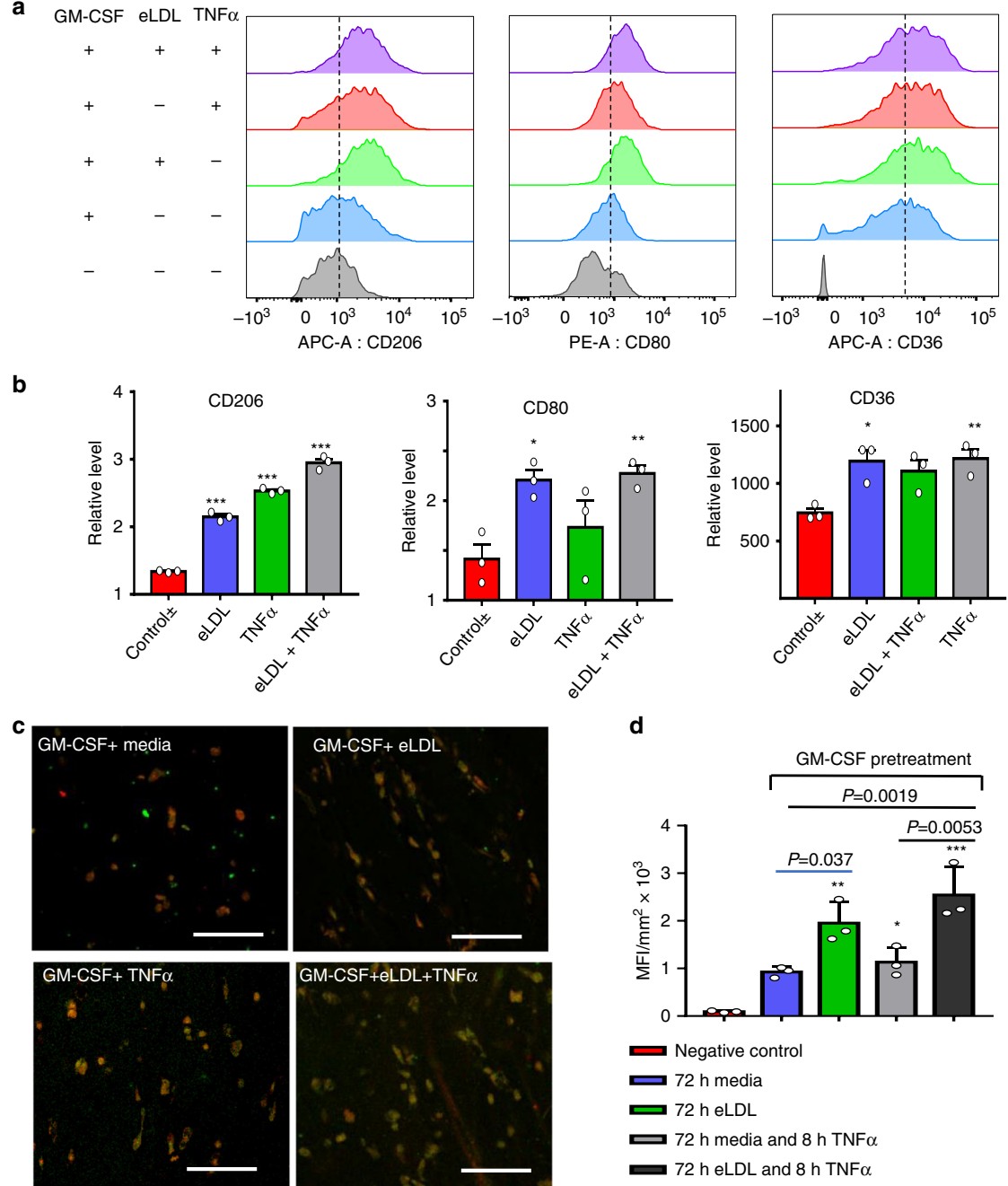

**Fig. 7 Macrophage polarization in TEBVs after exposure to eLDL and/or TNFα. a** Flow cytometry of primary human monocyte differentiation into macrophages with eLDL ± TNFα treatment. **b** Quantification of the mean fluorescence intensity (MFI) of flow cytometry results. The relative level represents normalization to negative control−/−(no GM-CSF/no treatments), (mean ± S.D., $n = 3$ independent wells from two monocyte isolations, CD206: ***$P <$ 0.0001; CD80: *$P = 0.0397$, **$P = 0.0267$; CD36: *$P = 0.0187$, **$P = 0.0143$, compared to control +/− =GM-CSF control/no treatments by one-way ANOVA and Tukey post hoc test). **c** Characterization of primary monocytes derived macrophages in perfused TEBVs with GM-CSF pretreatment and addition of eLDL for 96 h with or without 8 h TNFα treatment; fluorescence images of, monocytes-cell tracker red, CD80 + monocytes-green (scale bar 100 μm). **d** Quantification of mean fluorescence intensity (MFI) per cell of CD80 + monocytes (green signal) based on the images in panel (**c**) (mean ± S.D., $n = 3$ TEBVs, *$P = 0.034$, **$P = 0.00066$, ***$P < 0.0001$ compared to negative control by one-way ANOVA and Tukey post hoc test).

TEBVs were made and is used in both the fabrication mold and prefusion chamber. There are four steel hollow mandrels (outer diameter 0.63 mm, inner diameter 0.33 mm) at opposite sides of part A, which are mirror reflections of each other and link the assembled chamber to the perfusion tubing and pump. In the fabrication step, the halves of each mandrel are inserted into the chamber (Part A) and brought into contact with each other (Fig. S1ai and S1bi). Part B forms the top layer of the seeding mold with inlets/outlets and grooves, and part C is the bottom layer of the seeding mold with grooves (Suppl Fig. S1aii and S1bii). The grooves on the top and bottom layers each form four semicircular channels (diameter 2.2 mm, length 24

mm) to be used as molds. Once the high-density collagen containing the hNDFs is added and gelled, parts B and C are removed, and the collagen TEBVs are dehydrated. Then the mandrels are drawn out forming a lumen. Following fixation of the collagen tubing on the mandrels, two new flat covers are added (parts D and E) and the final perfusion chamber is completed. The steel mandrels are used for perfusion, with one side acting as the media inlet, and the other as the outlet. To fabricate the 3-layer TEBVs, parts A, F, G are first assembled to produce a mold with diameter 1.4 mm and length 24 mm (Fig. S1d). After seeding the mixture of hSMCs and collagen, the middle layer of TEBVs is allowed to gel. Then, parts F&G

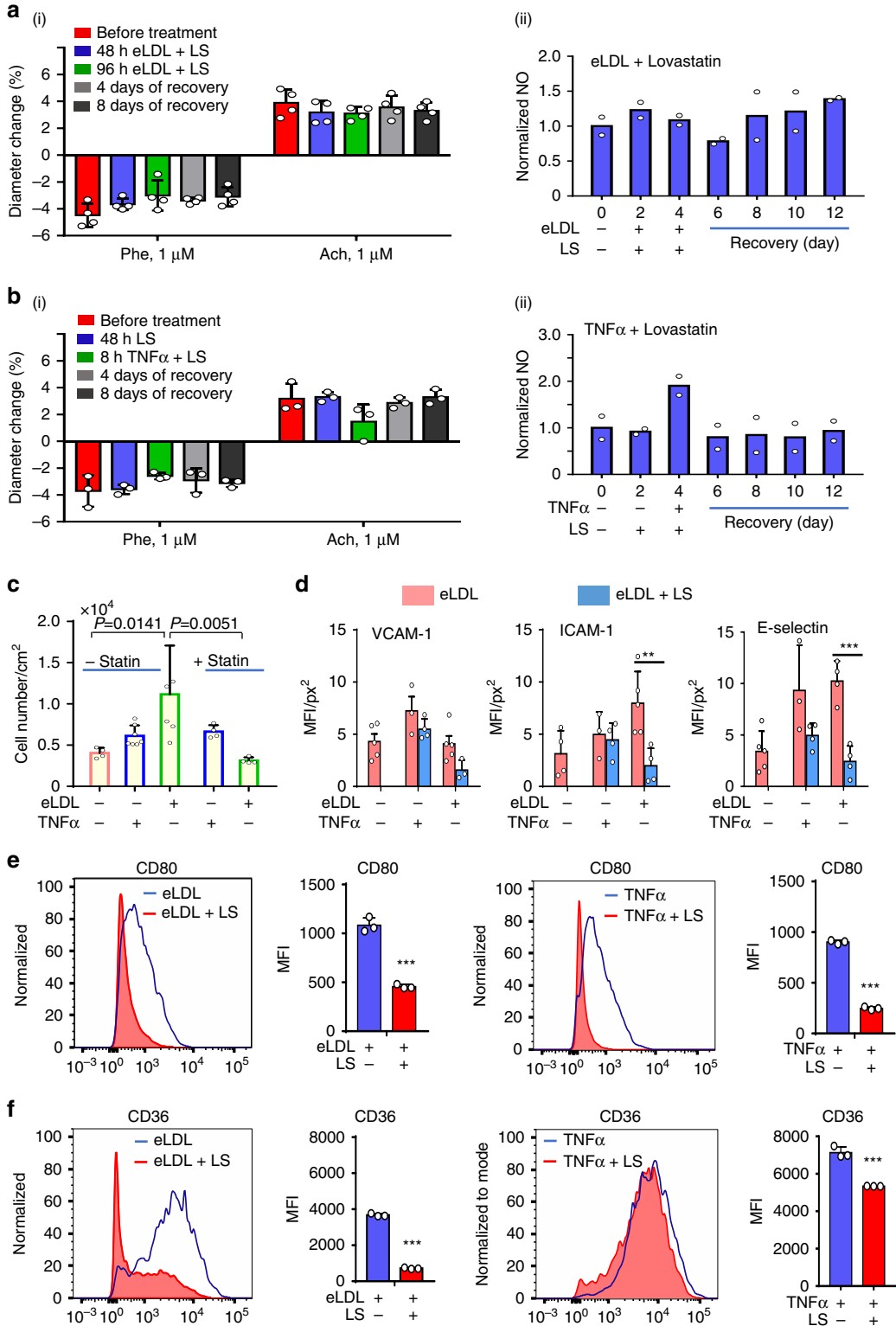

are removed and parts B&C are attached to part A to form the larger diameter mold as described above. Followed by seeding hNDFs and collagen mixture, the outer layer (diameter 2.9 mm) are formed. For both designs, after dehydration and suture of the TEBVs on mandrels, endothelial cells are perfused through the vessel lumen and allowed to adhere.

**Fabrication, endothelialization, and perfusion of TEBVs.** TEBVs were fabricated from collagen which underwent plastic compression to remove water and increase the mechanical strength[72]. High concentration rat tail collagen I (Corning, 8.2–9.4

mg/ml) was diluted to 7 mg/ml mixture on ice using 10 × Dulbecco's Modified Eagle's Medium (Sigma, final concentration 1 × in mixture), 1 M sodium hydroxide (Sigma, final pH 8.5) and the suspension of hNDFs (final cell density 10^6 cells /ml). The gel mixture was immediately injected into the seeding mold containing the four channels (180 μl/channel) and surrounding annular tubes which served as a mandrel (Fig. S1bi) and incubated at 37 °C for 30 min. The gelled collagen was dehydrated by exposing each side of the TEBVs to KimWipes (Whatman) 5 times. This removed about 90% of the water from the collagen gel (Fig. S2). Finally, the mandrels in the gels were drawn out 5–6 mm on each side to generate a channel in

**Fig. 8 Lovastatin blocks formation of early atherosclerosis in TEBVs.** Perfusion of 1 μM Lovastatin (LS) with 50 μg/ml eLDL blocks eLDL-mediated altered vasoactivity (**a–i**) (mean ± S.D., n = 4 TEBVs, Phe: phenylephrine, Ach: acetylcholine) and NO production (**a-ii**) in TEBVs (mean ± S.D., n = 2 from 8 TEBVs, R means recovery). **b** Lovastatin (1 μM) blocks the 50 U/ml TNFα mediated dysfunction of vasoactivity (i, mean ± S.D., n = 3 TEBVs, Phe: phenylephrine, Ach: acetylcholine) and NO production in TEBVs (ii, mean ± S.D., n = 2 from 8 TEBVs, R means recovery). **c** Lovastatin block eLDL-mediated monocyte accumulation in TEBVs (mean ± S.D., n = 4,7,5,4,4 TEBVs by two-way ANOVA). **d** Lovastatin inhibits ICAM-1 and E-selectin expression on endothelial cells in TEBVs (mean ± S.D., n = 5,3,4,5,3 TEBVs for VCAM-1 accordingly, n = 4,3,4,5,4 TEBVs for ICAM-1 accordingly, n = 5,3,4,4,4 TEBVs for E-Selectin accordingly, **P = 0.0097, ***P = 0.0007 by two-way ANOVA and Tukey post hoc test, MFI = mean fluorescence intensity). **e** Lovastatin (LS) blocks primary monocyte differentiation into macrophages, flow cytometry testing of CD80 (mean ± S.D., n = 3 independent wells from one monocyte isolations, ICAM-1: **P = 0.00016, E-Selectin: ***P < 0.0001 by Student's t test, MFI = mean fluorescence intensity). **f** Lovastatin (LS) blocks primary monocytes activation, flow cytometry testing of CD36 (mean ± S.D., n = 3 independent wells, ***P < 0.0001 by Student's t test, MFI = mean fluorescence intensity). (Flow rate: 0.5 ml/min per TEBV (0.4 Pa) for vasoactivity testing and 0.125 ml/min (0.1 Pa) for monocyte accumulation).

each TEBV. The ends of the newly generated vessels were tied using 4-0 silk sutures around the mandrel ends to eliminate leakage (Fig. S1avi). The hSMCs density in collagen was $10^6$ cells /ml. The seeding volume of hSMC/collagen mixture was 40 μl/channel and the seeding volume of hNDF/collagen mixture layer was 140 μl/channel.

The final culture chamber was assembled, and media was added to the chamber (Figs. S1av and S1c xiii). Next, 50 μl of an ECFC suspension ($8 \times 10^6$ cells/ml) was injected through the lumen of each TEBV. The chamber was sealed and rotated at 10 rotations/h for 45 min at 37 °C to allow for a uniform distribution of endothelial cells. Then the chip was attached to a flow loop containing a peristaltic pump (Masterflex) with a multi-channel pump head (Cole-Parmer) to create continuous flow through the four TEBVs in each chip. The flow rate of the pump was set to 0.5 ml/min per TEBV, which produced a shear stress of 0.4 Pa based on an average TEBV inner diameter of 647.5 ± 45.6 μm. For monocyte perfusion experiments, the flows of 0.5 ml/min per vessel (0.4 Pa) and 0.125 ml/min per vessel (0.1 Pa) were used to enable monocyte adhesion, which are comparable to the physiological shear stress[63]. The total perfusion volume was 3 ml in main vessel loop and 4 ml in the side loop (Fig. 1b). EGM media was used for the finished TEBVs and was changed every 2 days.

**Permeability of TEBVs to macromolecules.** Permeability assays were performed in acellular collagen constructs and in TEBVs. Fluorescein isothiocyanate (FITC) labeled goat IgG (Invitrogen) and 500 kDa FITC-dextran (Sigma) were diluted to 10 μg/ml and 20 μg/ml, respectively, in PBS and injected into the TEBV lumen. Serial fluorescence images with the same exposure time were taken every 5 min for up to 20 min. Then the permeability of TEBVs was calculated using equation[73]:

$$P = \frac{1}{\Delta I}\left(\frac{dI}{dt}\right)_{t=0}\frac{r}{2},$$

(1)

where $\Delta I$ is the change in total fluorescence intensity upon addition of labeled molecule ($t=0$)–Background signal; $\left(\frac{dI}{dt}\right)_{t=0}$ is the initial rate of transport of fluorescent dextran from the vessel lumen into the vessel, which can be approximated as $(dI/dt)_{t=0} \approx (I_t - I_{(t=0)})/t$, where $I_t$ is the intensity outside the vessel at time $t$, $I_{t=0}$ is the intensity outside the vessel at time $t=0$; $r$ is the radius of the lumen, which is estimated from width/2 of the fluorescent region at $t=0$. Based on published values for the hydraulic permeability of collagen gels undergoing plastic compression[74] ($10^{-14}$ $m^2$), the Peclet number for eLDL, which measures the relative value of the eLDL diffusion time to convection time is «1, indicating that the measured permeability is a diffusive permeability.

**Preparation of eLDL, TNFα, and Lovastatin solutions.** Aliquots of human plasma LDL were prepared by measuring the protein concentration of LDL (Lee Biosolutions) using the Lowry Protein Assay[75] and then diluting in a 10% sucrose (w/v) saline solution (150 mM NaCl, 0.24 mM EDTA, pH 7.4) to achieve a final concentration of 10 mg/ml. The sucrose enabled aliquots to be stored at −80 °C without loss of properties[75]. Enzyme modification of LDL was adapted from the method of Chellan et al.[46] 7 μg of trypsin (Sigma, T4049) and 12 μg of cholesterol esterase per mg LDL protein are added and incubated for 16 h at 37 °C. Next, 24 μg trypsin and 29 μg cholesterol esterase per mg of LDL protein are added and incubated for 48 h at 37 °C. In both cases, cholesterol esterase is administered 6 h after adding trypsin. After incubation, the eLDL solution is dialyzed again in PBS, pH 7.4 for 24 h. The eLDL is filter sterilized, its final protein concentration is measured, and stored at 4 °C. Particle size is characterized via dynamic light scattering (Malvern Zetasizer LS). LDL was oxidized by incubation with 10 μM $CuSO_4$ for 48 h[46].

TNFα (Sigma) was dissolved at a concentration of 10 μg/ml in sterile DPBS without calcium and magnesium. This concentration is equivalent to enzymatic activity of 200 U/μl. Stocks were kept frozen for 3 months before expiration and aliquots were used once with only one freeze thaw cycle. Lovastatin (Sigma) was dissolved in a solution of 40% ethanol (95–100%) and 60% 0.1 N NaOH. After heating at 50 °C for 2 h, the solution was neutralized with HCl to a pH of

approximately 7.2 and brought up to a volume of 1 ml with distilled water. The stock concentration of Lovastatin is 0.0198 mM.

**TEBV burst pressure, ultimate tensile stress, vasoactivity and nitric oxide (NO) production.** The burst pressure, ultimate tensile stress (UTS), vasoactivity, and NO production testing were preformed following established protocols in our lab[31]. Briefly, the burst pressure is tested in each TEBV with one end sealed, and an infusion of PBS at the other end until failure. Pressure was measured using a differential pressure gauge (Keller) and images were obtained to record the diameter. Then the ultimate tensile stress was calculated using the following equation:

$$UTS = P \times D/2t$$

(2)

where $P$ is the burst pressure, $D$ is the diameter of vessel lumen under burst pressure dilation, and $t$ is the thickness of the vessel wall that was calculated using ($D_{outer}$-$D_{lumen}$)/2 under burst pressure dilation. The lumen diameter, $D_{lumen}$, was determined using the value measured from histology for unpressurized vessels and the conservation of mass.

The vasoactivity of TEBVs was evaluated by placing the TEBV chip under a stereoscope (Amscope, software "ISCapture v3.6") at 9 × amplification, recording an image of the initial outer diameter, and adding 1 μM phenylephrine (Sigma) to the flow circuit. After 5 min, the outer diameter was recorded and acetylcholine (Sigma) was added to the perfusion media for 5 min at a final concentration of 1 μM to assess vasodilation. The outer diameter change of the TEBVs was analyzed using Image J (NIH). For NO production testing, the total concentration of nitrite and nitrate in the prefusion media was assessed using a Griess reagent assay kit (Pierce). Absorbance at 540 nm was measured using a Quant microplate reader (Bio-Tek). Samples were normalized to blank media controls.

**Monocytes perfusion, accumulation, and migration.** For monocyte perfusion, the TEBVs were perfused for one week to mature, then monocytes were added in the vessel loop only at $1 \times 10^6$ cells/ml for U937 cells and $2.5 \times 10^6$ cells /ml for primary monocytes. Total perfusion volume of the vessel loop was 3 ml. To measure monocyte accumulation and migration, monocytes were labeled with cell tracker red-CMTPX (Life Tech) before addition to the perfusion loop. At the end of monocyte perfusion, the TEBV lumen was washed with PBS gently to remove unattached cells. Then the TEBVs were fixed with 4% Paraformaldehyde (PFA) solution for one hour. The TEBV is opened en face and compressed between two coverslips. The accumulation of monocytes is analyzed based on fluorescence images and the migration of monocytes is analyzed from confocal images at different depths into the TEBV from the endothelial surface. For human whole blood perfusion, fresh collected umbilical cord blood with 25–30% citrate-phosphate-dextrose (CPD) is used to replace the perfusion media in vessel loop. $1 \times 10^6$ cells/ml of the cell tracker-CMTPX labeled U937 are added into whole blood and perfusion for 40 h. The side loop contained only media.

**Flow cytometry, immunofluorescence, and oil-red staining.** For flow cytometry, monocytes were stained with fluorescently labeled antibodies, including PE anti-human CD80 (Biolegend, 1:50), APC anti-human CD36 (Biolegend, 1:50), Pacific Blue™ anti-human CD206 (BD Pharmingen, 1:25) or Pacific Blue™ anti-human CD14 (Biolegend, 1:50), for 20 min on ice. The APC Mouse IgG2a (Biolegend, 1:50), PE Mouse IgG1(Biolegend, 1:50), and Pacific Blue™ Mouse IgG1 (Biolegend. 1:50) were used to stain the cells as isotype control, which was used to set the selected gate during testing. Then stained cells were collected by FACSCanto (BD Bioscience) and analyzed by FlowJo (Treestar, MAC version, v9.3.2) software. The gating strategy is shown in Fig. S11. For immunostaining, TEBVs are perfused and gently washed using PBS then fixed in 4% paraformaldehyde for 60 min at room temperature. Following fixation, samples are washed in PBS then blocked in blocking buffer (PBS + 10% goat serum+5% BSA) for 2 hr at room temperature. Cross-sections are obtained on a cryostat (Leica CM3050). The following primary antibodies were used for vessel characterization: CD31 (BD, 1:200), α-smooth muscle actin (αSMA, Abcam, 1:150), smooth muscle myosin heavy chain 11 (MHC11, Abcam, 1:100), CD80

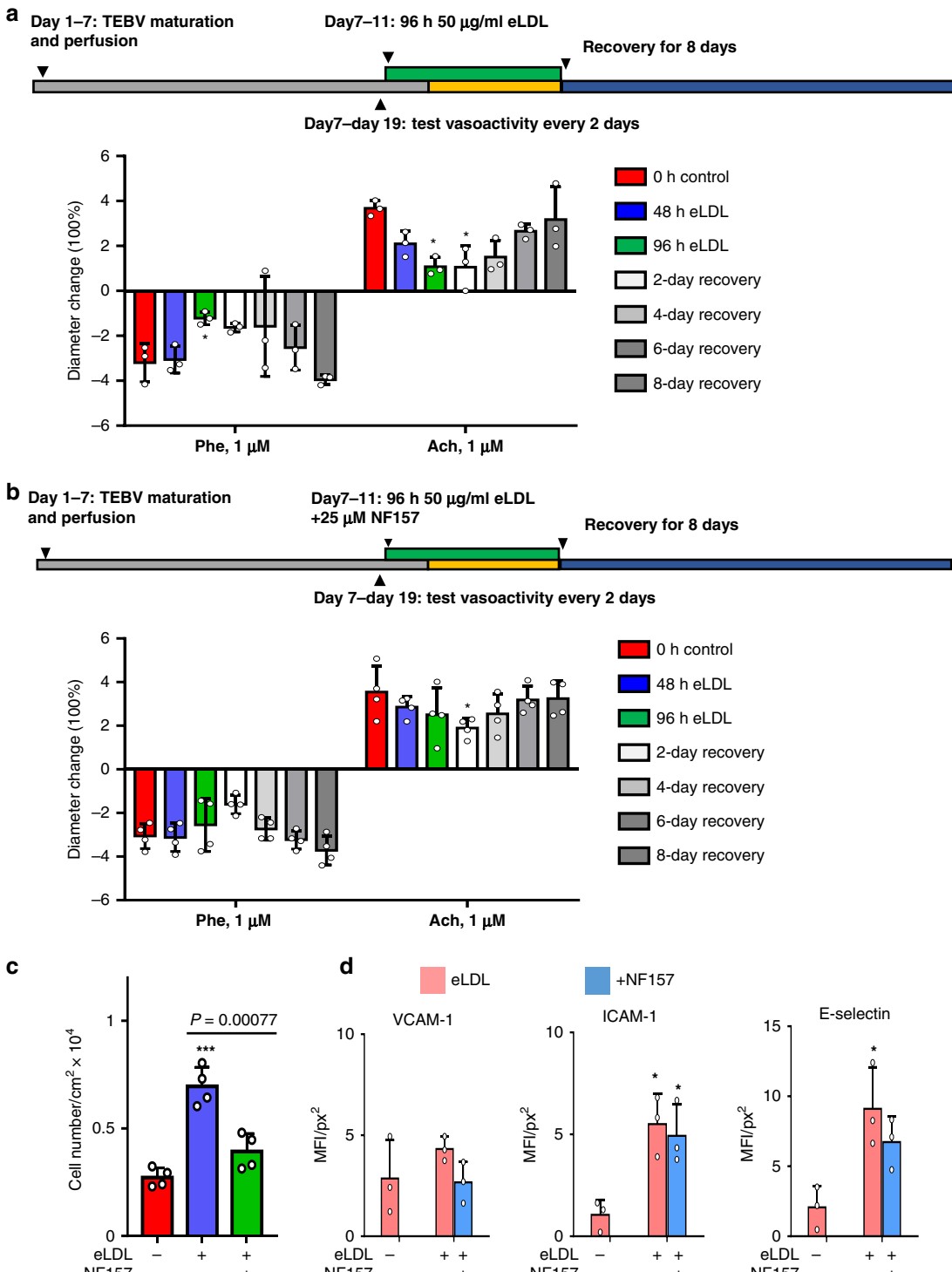

**Fig. 9 NF157 block events in early atherosclerosis in TEBVs. a** Perfusion of TEBVs with 50 μg/ml eLDL reduces vasoconstriction and vasodilation, but the response recovers 8 days after removal of eLDL (mean ± S.D., $n = 4$ TEBVs, *PE: $P = 0.01408$, Ach: *$P = 0.01543$ to control by two-way ANOVA and Tukey post hoc test). **b** Addition of 25 μM NF157 with 50 μg/ml reduces the inhibition of vasoactivity and leads to a more rapid recovery after removal of eLDL (mean ± S.D., $n = 4$ TEBVs, *$P = 0.01369$ to control by two-way ANOVA and Tukey post hoc test). **c** NF157 block eLDL-mediated monocyte accumulation in TEBVs (mean ± S. D., $n = 4$ TEBVs, *$P < 0.05$, ***$P < 0.0001$ to control by two-way ANOVA). **d** NF157 does not inhibit ICAM-1 and E-selectin expression on endothelial cells in TEBVs (mean ± S.D., $n = 3$ TEBVs, ICAM-1: *$P = 0.0136$ control vs. eLDL, *$P = 0.0252$ control vs. eLDL +NF157; E-selectin *$P = 0.0181$ control vs. eLDL by two-way ANOVA and Tukey post hoc test, MFI = mean fluorescence intensity; Flow rate: 0.4 Pa to vasoactivity testing and 0.1 Pa for monocyte accumulation).

(BioLegend, 1:100), VCAM-1 (Santa Cruz Biotechnology, 1:200), ICAM-1 (Santa Cruz Biotechnology, 1:200), E-selectin (Santa Cruz Biotechnology, 1:200). Then the primary antibodies were treated with fluorescently labeled secondary antibodies: 488-goat anti mouse IgG, 488-goat anti rabbit IgG, 546-goat anti mouse IgG, 546-goat anti rabbit IgG, 633-goat anti mouse (1:200) (Life Technologies)

and Hoechst dye (1:1000) to stain nuclei. Images were acquired using a Leica SP5 inverted confocal microscope (software "LAS AF v2.7.3.9723"). Oil Red staining was performed by using standard protocols on vessels fixed in 4% paraformaldehyde. The images were acquired using a fluorescence microscope (Nikon model TE2000-U, software "NIS-Elementa AR v3.2").

**RNA isolation and reverse transcriptase-PCR analysis**. The primer sequences (Table S1) are obtained from PubMed and are generated using the online design program Primer-Blast. Total RNA was isolated from the entire TEBV directly using RNeasy Fibrous Tissue Mini Kit (QIAGEN). The purity and concentration of RNA were measured using a Nano-Drop Spectrophotometer. For reverse transcription of RNA into cDNA, 250 ng RNA from the entire TEBV is used for each sample. Reverse transcription was performed using the iScript cDNA Synthesis Kit (BioRad). The PCR reaction was mixed by following the protocol of the iQ SYBR Green Supermix (Bio-Rad). RT-PCR was performed using a CFX Connect Real-Time PCR Detection System (Bio-Rad).

**Statistics and reproducibility**. Data were analyzed using Microsoft Excel, Originlab, and Graphpad Prism. Results are presented as mean ± S.D. unless noted otherwise. Statistical significance is determined by Student's $t$ test between two groups and ANOVA with a post hoc Tukey test for multiple groups. A value of $p < 0.05$ was considered statistically significant. All the images were confirmed or observed for at least 3 separate experiments in this work.

**Reporting summary**. Further information on research design is available in the Nature Research Reporting Summary linked to this article.

## Data availability

The authors declare that all data supporting the findings of this study presented in the graphs are available in Source Data file and other results or can be requested from the authors. Source data are provided with this paper.

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

## Acknowledgements

This work was supported by NIH Grants UG3TR002142 and UH3TR002142 from NIAMS and NCATS, and R01HL138252. M.B. is supported on an NIH 1R38HL143612-01 fellowship. J.J.B. was supported by a fellowship from the Sarnoff Cardiovascular Research Foundation.

## Author contributions

X.Z. and G.A.T. designed the project. X.Z. and G.A.T. developed the device. X.Z., G.Z., V.P., E.S., and J.J.B. performed the experiments and analysis. E.S. and Q.Z. isolated the ECs from blood. X.Z., M.B., and G.A.T. analyzed the data and co-wrote and edited the manuscript.

## Competing interests

The authors declare no competing interests.
