## [Peer Review File · Nature Communications]

Reviewers' comments:

Reviewer #1, an expert in atherosclerosis (Remarks to the Author):

Zhang et al. have used primary human endothelial colony-forming cells and primary human neonatal dermal fibroblasts (hNDF) and a novel fabrication method to create an in vitro model of arterioles. They have systematically characterized this model and used it as a tool to study early stages of atherosclerosis. Overall, I think this is a very interesting model of blood vessels that can be adopted and used for different purposes.

The manuscript is well written and the data are well presented and discussed. Although this reviewer acknowledges the sophistication of the model developed, it is not clear what insight this model actually provides. It seems the authors present strong technical achievements without really reporting on novel mechanisms or novel insights. This is a major limitation of this manuscript, which dampens the overall enthusiasm for this work. Most of the data obtained could be achieved by standard cell culture experiments. The authors did not use the advantage of having a flow model in regards to changing flow rates, shear stress of imitating stenoses. Such experiments would build on the advantages of the authors microfluidic system. As such the authors should invest in further work and try to use their (without any doubt highly sophisticated) model for functional investigations aiming to provide novel insights/mechanisms.

I have the following specific comments:

In the authors' model a layer of smooth muscle cells is missing. The authors need to add this limitation to their discussion or even better present some data on including smooth muscle cells in their approach.

I recommend the author to include the schematic image or a photo of the setting during their experiments, TEBV, pump, incubator etc

The authors should discuss the appropriateness of the concentrations of eLDL used.

Perfusion of TEBVs with media for 1-4 weeks. When do the authors change the media and what is the basis of the time points chosen?

Page 22, primary monocyte differentiation into macrophages: It is not clear how the experiments have been conducted. For how long did the monocytes stay in circulation inside the TEBV model. What was the shear stress level during these experiments?

I found plenty of Typos in the manuscript. The authors should do or send the manuscript for proper proofreading.

(Fig. 3gv) should change to (Fig. 3g-v)

the permeability of TEBVs was still $2.1 \pm 1.3 \times 10^{-6}$ at 10min add space, 10 min and many more.

In general figures should have a consistent lay out (e.g. either use bold or not throughout the figures, some have sublabels i, ii,iii other don't)

Reviewer #2, an expert in microfluidic analysis of immune cells (Remarks to the Author):

This manuscript describes the creation and validation of an in vitro tissue-engineered blood vessel (TEBV) model, and its utilization to model several key aspects of early stage atherosclerosis. The TEBV consists of 4 parallel arteriole-scale straight tubes fabricated using fibroblasts-embedded

Collagen I with an inner lumen lined by endothelial cells, exhibiting in vivo-like vasoactivity, endothelial release of nitric oxide, and good barrier function. Upon stimulation by eLDL, the TEBV showed several key features of early pathological events in atherosclerosis, including EC inflammation, impaired vasoactivity and barrier function, increased monocyte accumulation and foam cell formation of EC, monocyte, and fibroblast. Lastly, it was demonstrated monocyte accumulation and foam cell formation caused by exposure to eLDL can be reduced by treatment of lovastatin.

Overall the manuscript is well written, and results are well organized. Technical novelty is low (see below), and my concern is on the limited use of stimulus (eLDL and TNF α) and cell lines, and the lack for whole blood perfusion in the developed model. The paper can be further strengthened if other clinically relevant conditions (e.g. high glucose) and pathological shear stresses can be shown to induce similar EC and monocyte pro-inflammatory phenotype.

While the studies on effects of eLDL for each cell type are quite elaborate and extensive, there is a lack of methodological novelty for the TEBV model. Recapitulation of physiological vascular function have been previously reported by a number of groups, but the author failed to cite any of them. Joe Tien's group had reported a vascular model fabricated with a very similar method as this study (Chrobak et al. 2006, Microvascular research). The vascular model reported by Abraham D. Stroock's groups (Zheng et al, 2012 PNAS) also showed high level recapitulation of vascular function which was fabricated using a similar method too. Introduction should include these work, and more recent atherosclerosis-on-a-chip microfluidic technologies (Menon et al, 2017 Lab Chip, Zheng et al, 2016 Small).

Engineering: Is the removal of 90% of water from collagen gel physiological? Since the chip platform is small and vessels are separated by collagen, are there any cross talks (molecular diffusion) between the vessels?

The flow rate used for monocyte perfusion generate a shear stress (0.17Pa) which is far lower than the physiological shear stress of artery. The author mentioned that this was to promote monocyte accumulation, which means the accumulation observed in the present study is not a good representation of in vivo condition. This should be justified and discussed. It will be good if an additional study can be done to characterize the upper limit of shear stress that the system can withstand, and monocyte accumulation can still be observed. Recent in vitro blood vessel model also reported whole blood perfusion to study leukocyte-endothelial interactions (Menon et al, 2017 Lab Chip). This should be demonstrated in this study.

While a co-culture model was used here, there are very little results on the perivascular cells. It will be important to characterize the phenotypes in the presence and absence of fibroblast. Secondly, fibroblasts were used to create the vessel wall of arteriole. It would be more physiologically relevant to use vascular SMC which is the major cell component in the medial layer and also plays an important role in disease progression of atherosclerosis. Several studies were based on the vasoactivity of the TEBV, which is characterized by changes in vessel lumen diameter, however the main cell type responsible for contraction and relaxation is SMC. In the study where the fibroblast is contracting/relaxing, it may not be a good representation of in vivo condition. The author should justify and discuss the usage of fibroblast in their model. Besides, CFEC was used to create the endothelium in the TEBV to model arterial biology. The cell biology of CFEC can be quite different from its fully matured counterpart. It would be good if an additional study can be done to valid that the findings can be replicated in arterial EC.

Figure 2: TEBV stability up to 4 weeks were demonstrated by maintenance of burst pressure, NO production and response to Phe and Ach. However, it is still necessary to include an image showing the EC monolayer is intact and positive of CD31, as it is very rare that confluent EC monolayer can last that long in previous reports.

Figure 3g: Images are too small to read. Suggest to move some to SI. From the brightfield image,

it seemed that the vessel diameter and wall thickness (fibroblast-dependent) varied significantly between different conditions. These two factors can affect vessel permeability too. For permeability results to be valid, comparable lumen size and vessel wall thickness should be chosen for different conditions. Additionally, one of the highlights of the TEBV is the perfusion culture, but the author failed to show any improvement on cell phenotype of perfusion culture over static culture and perfused condition. It will be good if vessel wall permeability of static culture can also be included in this dataset.

Figure 3g, 3v: As described in Methods section, the permeability of the EC barrier is calculated using the equation derived from Fick's first law, which is first described by Huxley et al. 1987. (Am J Physiol.) In the derivation of the equation, $(dI/dt)_0$ is the initial increase in fluorescent intensity as solute begins to diffuse into the field. For the equation to be valid, the fluorescent intensity change should be captured immediately after the introduction of solute into the vessel, which is usually done within 3 min as reported by other groups. In this study, the time points used was 10min and 20min, which can't be considered as initial increase. The author should capture the changes within 3 min, quantify $(dI/dt)_0$ by fitting a linear trendline over several timepoints, and use the value of gradient for $(dI/dt)_0$ rather than obtaining multiple values for different timepoints.

Figure 4c: There are limited results on the ECM re-modeling section. ECM secretion is characterized using qRT-PCR, but there is no indication of any internal control/housekeeping gene (e.g. GAPDH) for all datasets. For qRT-PCR results to be valid, at least one internal control gene should be included as the baseline to normalize the CT value of each sample. Additionally, ECM remodeling is hallmark of atherosclerosis progression, although it was shown ECM mRNA was upregulated after 8 days recovery, actual secretion of these ECM components could be validated by immunostaining.

Figure 6a, 6b. Conditions with eLDL only/TNF α only should also be included for both datasets to prove whether LS blocked the drop in vasoactivity induced by eLDL or TNF α

IL-1 β and IL-1 were used interchangeably in the manuscript, the author should make these consistent.

Reviewer #3, an expert in in vitro blood vessel models (Remarks to the Author):

They created in vitro model of early stage atherosclerosis by fabricating, endothelializing and perfusing arteriole-scale tissue-engineered blood vessels (TEBVs).

The tube which is made by human neonatal dermal fibroblasts (hNDF) and collagen gel and endothelialized in inner side using primary human endothelial colony forming cells (ECFCs). This TEBVs is useful model and have advantage (overcoming the lack of appropriate methods of the results to human cases and the expensive & time consuming) to examine the effect of genetic variants and pathogenesis of human vascular diseases.

However, there are some concerns about the cellular component for TEBVs and interpretation on the results.

(1) They demonstrated the endothelial multilayered TEBVs replicate arteriole-mimic function, e.g. vasoreaction to phenylephrine and endothelial-dependent vasodilation. According to this evidence, hNDF acts as vascular smooth muscle cells (VSMCs). It is better to reveal the characterization of hNDF, i.e. expression of key molecules to mediate constructive and relaxation response through α -adrenergic receptor, NO-GC respectively. Even if so, VSMCs are more ideal cell components in TEBVs. Why did they utilize fibroblasts instead of VSMCs in TEBVs?

(2) They also observed that adherent and accumulation of inflammatory cells and transformation of macrophages from circulating monocytes within the walls of TEBVs. According to their results, as compared to 2D-culture system, the phenomenon is dependent on the reaction between

inflammatory cells and endothelium (EC). Please explain or demonstrate the advantage to use multilayered TEBVs instead of the EC-made perfusion tubes.

(3) Mechanical factors such as shear stress are crucial for the pathogenesis of atherosclerosis. In this point, the perfusion system of TEBVs has advantage. However, the tubular size (less than 1mm) of TEBVs is far different compared to human atherosclerotic vessels such as coronary artery (2-5 mm in diameter). Please explain the substitutional points and the scale-based limitation between TEBVs and actual human vessels.

Minor comments

- (1) In figure 1, it is hard to distinguish the colors indicating fibroblast and eLDL-activated fibroblasts. It is better to consider the color design for being kind to color-blind readers
- (2) In figure 3 (g, h), what does it compare to? Please indicate the number of explanation.
- (3) In figure 4 legend, TNF α may be fixed to TNF α (alpha).
- (4) In figure 5 (k, l), hard to follow up the results. In figure 5k, solely green-signals are observed. CD81+ monocytes (green) are detected within total red-labeled monocytes? MFI of double positive cells are measured?

Overall Changes

Changes to the text and figure legends are denoted in red. We divided several figures so that the images are larger. New results to the manuscript include:

1. Comparison of two- and three TEBVs (Figs. 1g, 2g, S1c,d, S3, s6);
2. Endothelial orientation in TEBVs (Figs. 2f, S4)
3. Monocyte adhesion and EC activation at two flow rates (Figs. 5a,b, 6f)
4. Effect of whole blood on monocyte adhesion (Figs. 6i, S8)
5. Response of TEBVs to NF157 with eLDL treatment (Fig. 9).

Reviewer #1, an expert in atherosclerosis (Remarks to the Author):

Zhang et al. have used primary human endothelial colony-forming cells and primary human neonatal dermal fibroblasts (hNDF) and a novel fabrication method to create an in vitro model of arterioles. They have systematically characterized this model and used it as a tool to study early stages of atherosclerosis. Overall, I think this is a very interesting model of blood vessels that can be adopted and used for different purposes.

Response: Thank you very much for the positive comments.

1. The manuscript is well written and the data are well presented and discussed. Although this reviewer acknowledges the sophistication of the model developed, it is not clear what insight this model actually provides. It seems the authors present strong technical achievements without really reporting on novel mechanisms or novel insights. This is a major limitation of this manuscript, which dampens the overall enthusiasm for this work. Most of the data obtained could be achieved by standard cell culture experiments. The authors did not use the advantage of having a flow model in regards to changing flow rates, shear stress of imitating stenoses. Such experiments would build on the advantages of the authors microfluidic system. As such the authors should invest in further work and try to use their (without any doubt highly sophisticated) model for functional investigations aiming to provide novel insights/mechanisms.

RESPONSE: Thank you for these points. While several measurements could be obtained with isolated cells (e.g. endothelial permeability and endothelial orientation after exposure to shear stress), vasoactivity and monocyte accumulation and foam cell formation can only be obtained with the three-dimensional engineering blood vessel system. Further, key endothelial functions, such as permeability are influenced by EC-SMC interactions. A key measure of vascular function is vasoactivity which is one of the earliest changes in atherosclerosis, signaling endothelial dysfunction. This measure can only be obtained with an entire 3D vessel mimic.

Specifically, in this work, we showed a novel response of eLDL on TEBV vasoconstriction over a period of several weeks and that enzyme-modified LDL inhibits vasodilation. (Such an experiment is challenging to perform with isolated vessels, which can only be maintained for a few hours.) We were able to examine whether vessel function and atherogenesis were reversed when modified LDL was removed. Most importantly, we examined how two different drugs influenced foam cell formation in TEBVs. These studies demonstrate the utility of the system to test drugs that have both systemic and local effects.

As the reviewer's suggestion, we studied monocytes accumulation and EC activation under two different flow rates of 0.125 ml/min and 0.5 ml/min which produced shear stresses of 0.1 Pa and 0.4 Pa, respectively (Fig. 6f). We also examined the effect of whole blood on monocyte adhesion in our system. (Fig. 6i).

Reviewer #1 Specific Comments:

1. In the authors' model a layer of smooth muscle cells is missing. The authors need to add this limitation to their discussion or even better present some data on including smooth muscle cells in their approach.

RESPONSE: We had not included smooth muscle cells previously due to our prior experience and other reports that cultured primary human smooth muscle cells have reduced contractility. As a result, we used human neonatal dermal fibroblasts which exhibited significant contractility and adopted a myofibroblast phenotype. To improve the model though and assess whether inclusion of smooth muscle cells would improve contractility, we successfully fabricated the TEBVs with 3 layers: endothelial layer, human coronary artery smooth muscle layer, and human neonatal dermal fibroblast layer. We also compared the vasoactivity function of these 3-layer TEBVs to 2-layer ones with fibroblasts or SMCs. The results shown that the vasoactivity and burst pressure of 3-layer TEBVs and 2-layer TEBVs with fibroblasts are better than TEBVs with SMCs (Fig. 1g, Fig. 2g and Fig. S6).

2. I recommend the author to include the schematic image or a photo of the setting during their experiments, TEBV, pump, incubator etc

RESPONSE: Thank you for the suggestion. We added a photo of the perfusion setting in supplemental materials (Fig. S8).

3. The authors should discuss the appropriateness of the concentrations of eLDL used.

RESPONSE: The reason we used 50 $\mu\text{g}/\text{mL}$ eLDL was that we found the longer exposure (more than 48h) of 100 $\mu\text{g}/\text{mL}$ eLDL causes the ECs to detach whereas ECs tolerated a 96 h exposure 50 $\mu\text{g}/\text{mL}$ eLDL. Besides, 50 $\mu\text{g}/\text{mL}$ eLDL exposure induced foam cells generation to both monocytes and fibroblasts. Thus, we used 50 $\mu\text{g}/\text{mL}$ eLDL to treat the TEBVs. We added this point in the manuscript (Fig.6b). Further, this concentration of eLDL is similar to that used in other studies with eLDL (ref. 55, 61) or oxidized LDL (ref. 59). We note this in the manuscript.

4. Perfusion of TEBVs with media for 1-4 weeks. When do the authors change the media and what is the basis of the time points chosen?

RESPONSE: We change the media in both the vessel loop and side loop every other day. We chose this time point because there were about 0.7M-0.9M cells totally in one chip (about half number of cells in one T75 flask). The frequency was chosen based on the frequency of media changes for similar cell numbers and media volumes in 2D culture.

5. Page 22, primary monocyte differentiation into macrophages: It is not clear how the experiments have been conducted. For how long did the monocytes stay in circulation inside the TEBV model. What was the shear stress level during these experiments?

RESPONSE: This is a good point. The flowrate of perfusion is 0.125 ml/min (0.1Pa) per TEBV or 0.5 ml/min per chip (0.4 Pa) and the monocytes were perfused for 72 h. The higher flow rate conditions (0.4 pa) corresponded to 2 ml/min per chip. We added this information in the manuscript and figure legends.

6. I found plenty of Typos in the manuscript. The authors should do or send the manuscript for proper proofreading.

(Fig. 3gv) should change to (Fig. 3g-v)

the permeability of TEBVs was still $2.1 \pm 1.3 \times 10^{-6}$ at 10min add space, 10 min and many more.

In general figures should have a consistent lay out (e.g. either use bold or not throughout the figures, some have sublabels i, ii,iii other don't)

RESPONSE: Thank you for the comments. We corrected the errors noted and checked the entire manuscript again and corrected the typographical errors. We reformatted the figures to be consistent throughout.

Reviewer #2, an expert in microfluidic analysis of immune cells (Remarks to the Author):

This manuscript describes the creation and validation of an in vitro tissue-engineered blood vessel (TEBV) model, and its utilization to model several key aspects of early stage atherosclerosis. The TEBV consists of 4 parallel arteriole-scale straight tubes fabricated using fibroblasts-embedded Collagen I with an inner lumen lined by endothelial cells, exhibiting in vivo-like vasoactivity, endothelial release of nitric oxide, and good barrier function. Upon stimulation by eLDL, the TEBV showed several key features of early pathological events in atherosclerosis, including EC inflammation, impaired vasoactivity and barrier function, increased monocyte accumulation and foam cell formation of EC, monocyte, and fibroblast. Lastly, it was demonstrated monocyte accumulation and foam cell formation caused by exposure to eLDL can be reduced by treatment of lovastatin.

RESPONSE: Thank you very much for the comments.

1. Overall the manuscript is well written, and results are well organized. Technical novelty is low (see below), and my concern is on the limited use of stimulus (eLDL and TNF α) and cell lines, and the lack for whole blood perfusion in the developed model. The paper can be further strengthened if other clinically relevant conditions (e.g. high glucose) and pathological shear stresses can be shown to induce similar EC and monocyte pro-inflammatory phenotype.

RESPONSE: Thank you for the points. Based on the suggestions, we examine the effect of perfusion with whole blood in our model (Fig. 6 h-i), and tested monocytes accumulation and EC activation under higher shear stresses (Fig.5 a-b and Fig.6f). Furthermore, we successfully fabricated the TEBVs with 3 layers in added some related results in this version (Fig. 1g, Fig. 2g and Fig. S6).

2. While the studies on effects of eLDL for each cell type are quite elaborate and extensive, there is a lack of methodological novelty for the TEBV model. Recapitulation of physiological vascular function have been previously reported by a number of groups, but the author failed to cite any of them. Joe Tien's group had reported a vascular model fabricated with a very similar method as this study (Chrobak et al. 2006, Microvascular research). The vascular model reported by Abraham D. Stroock's groups (Zheng et al, 2012 PNAS) also showed high level recapitulation of vascular function which was fabricated using a similar method too. Introduction should include these work, and more recent atherosclerosis-on-a-chip microfluidic technologies (Menon et al, 2017 Lab Chip, Zheng et al, 2016 Small).

RESPONSE: There are two novel technical features in our work: 1) a chip to rapidly fabricate and perfuse four arteriole-scale vessels, and 2) modeling the early stages of atherosclerosis. Most prior studies focused on larger TEBVs (e.g. 6000 μm ID) for implantation, a lengthy process taking 8-12 weeks. We developed an approach that could produce four TEBVs (647.5 ± 45.6 μm ID) in a single chip, perfuse them within 24 h and use within a week of fabrication. This allowed us to greatly increase throughput of our system. While other studies examined one or two aspects related to atherosclerosis (e.g. inflammation, or thrombosis), we examined many of the early steps including activation of endothelium with modified LDL and cytokines, monocyte attachment, transmigration and accumulation and foam cell formation. We are not aware of other *in vitro* models that reproduced all of these features of the disease. These points are elaborated by comparing with the published studies noted above.

Omitting the references cited by the reviewer was an oversight on our part. However, these very fine papers had different goals than our study. Zheng et al. 2012 focused on modeling capillary networks. They did examine the role of perivascular cells in angiogenesis and examined blood endothelial cell interactions. Zheng et al. 2016 cultured an endothelial layer on a deformable substrate but did not include smooth muscle cells. They examined effect of shear stress and mechanical deformation of the vessel mimic. They separately examined high glucose or cholesterol on reactive oxygen species formation and LPS on inflammation but did not reconstruct the detailed steps in early atherosclerosis that we did. The strength of the Menon et al. papers was the characterization of EC function and examination of different geometries. They examined effect of SMCs on EC permeability but did not consider vasoactivity. To mimic atherosclerosis, they examined the response of endothelium, a stenosis, but did not consider monocyte accumulation and foam cell formation.

Our study differed in several ways from these other studies. We considered endothelial cell-smooth muscle cell interactions on a key functional measure, vasoactivity. This was chosen because altered vasoactivity is one of the earliest manifestations in atherosclerosis. Thus, it was necessary to create a separate vessel with multiple cell types. We also modeled the key steps of monocyte adhesion, transmigration and monocyte and smooth muscle foam cell formation which has not been done before *in vitro*. We report novel results about the effect of modified LDL on vasoconstriction. Finally, we examined how two different drugs affected these events. None of these experiments were performed in the works cited above.

We added a paragraph to the Introduction describing the different approaches to model the vasculature *in vitro* and cited the related papers of each strategies. The comparison among different strategies could help the readers to understand better how the vascular MPS we developed differed from other work in the literature.

3. Engineering: Is the removal of 90% of water from collagen gel physiological? Since the chip platform is small and vessels are separated by collagen, are there any cross talks (molecular diffusion) between the vessels?

RESPONSE: Yes, the removal of 90% of water from collagen gel creates a more physiological situation and is very crucial to make TEBVs with collagen. The collagen density of *in vivo* blood vessels is in the range of 70 mg/g-120mg/g \approx 70mg/ml-120mg/ml). In this work, the density of initial collagen tubing was 7 mg/ml. After dehydration, the density rose to 70 mg/ml, which was very similar to native vessels and led to a significant increase in the TEBV mechanical strength.

To the second question, we want to clarify that the collagen matrix is part of each TEBV, and the TEBVs are separated by culture media which is perfused through side ports, as shown schematically in Figure 1b.

The collagen with fibroblasts (or fibroblasts+SMC) is the vascular wall, accordingly, the EC layer is only the surface exposed to shear stress.

Cross talk may occur between vessels in the same chamber. The vascular wall is 200 μm thick and small molecules can diffuse that distance in a few minutes while proteins may take about 30 minutes to diffuse across the vessel wall. The media is changed every other day allowing accumulation of metabolites, modified LDL, or cytokines generated. The result is that this may create uniformity among the 4 TEBVs in a chip and is one reason that all TEBVs in a chip are exposed to the same conditions.

4. The flow rate used for monocyte perfusion generate a shear stress (0.17 Pa) which is far lower than the physiological shear stress of artery. The author mentioned that this was to promote monocyte accumulation, which means the accumulation observed in the present study is not a good representation of in vivo condition. This should be justified and discussed. It will be good if an additional study can be done to characterize the upper limit of shear stress that the system can withstand, and monocyte accumulation can still be observed. Recent in vitro blood vessel model also reported whole blood perfusion to study leukocyte-endothelial interactions (Menon et al, 2017 Lab Chip). This should be demonstrated in this study.

RESPONSE: For monocyte perfusion, the flow rate was decreased to 0.5 ml/min per chamber (0.1 Pa in each vessel) and the shear stress was lower than physiological shear stress of artery in vivo (0.3-1.3 Pa). We chose this lower shear stress to model low shear stresses at arterial branches (ref. 66) and to ensure that monocytes would adhere. As the reviewer suggested, we further tested the monocytes accumulation under 2 ml/min per chamber. In this way, the shear stress is 0.4 Pa in each vessel and more physiological. Comparing the results between lower and higher shear stress (Fig. 6f), we found the higher shear stress only decreased monocyte adhesion in control (untreated) group but did not affect monocyte accumulation in all treatment groups.

The upper limit of flow rate in our system depended on the fabrication quality of the acrylic device, rather than the TEBVs which could bear 1 bar burst pressure. At present, the highest flow rate is 2 ml/min per chamber now to avoid leakage. We are examining ways to further reduce the TEBV diameter which will have the most significant effect on shear stress without further increasing flow rate.

5. While a co-culture model was used here, there are very little results on the perivascular cells. It will be important to characterize the phenotypes in the presence and absence of fibroblast. Secondly, fibroblasts were used to create the vessel wall of arteriole. It would be more physiologically relevant to use vascular SMC which is the major cell component in the medial layer and also plays an important role in disease progression of atherosclerosis. Several studies were based on the vasoactivity of the TEBV, which is characterized by changes in vessel lumen diameter, however the main cell type responsible for contraction and relaxation is SMC. In the study where the fibroblast is contracting/relaxing, it may not be a good representation of in vivo condition. The author should justify and discuss the usage of fibroblast in their model. Besides, CFEC was used to create the endothelium in the TEBV to model arterial biology. The cell biology of CFEC can be quite different from its fully matured counterpart. It would be good if an additional study can be done to valid that the findings can be replicated in arterial EC.

RESPONSE: As we now note in the text, we chose human neonatal dermal fibroblasts rather than primary vascular smooth muscle cells due to the limited contractility of these SMCs (ref. 57 and 58). In the revised version, we successfully fabricated the TEBVs with 3 layers: endothelial layer, smooth muscle

layer, and fibroblasts layer. Staining of SMCs fibroblasts show that while the SMCs occupied the inner region of the TEBVs, both SMCs and fibroblasts were positive for α -smooth muscle actin (Fig. 1g).

Immunohistochemical staining (Fig. S3) shows the presence of α smooth muscle actin and the late differentiation smooth muscle myosin heavy chain (MHC11). Furthermore, we compared the vasoactivity and burst pressure of these 3-layer TEBVs to 2-layer ones with fibroblasts or SMCs to reveal the role of fibroblasts and SMCs in vessels. Our results do confirm that the fibroblasts are more contractile than smooth muscle cells.

We also recently published a model using iPS-derived endothelial cells and smooth muscle cells to model the rare genetic disease progeria using a single TEBV chamber (Stem Cell Reports **14**, 325-337 (2020)). These results do suggest that iPS-derived SMCs from healthy donors are more contractile than primary SMC.

6. Figure 2: TEBV stability up to 4 weeks were demonstrated by maintenance of burst pressure, NO production and response to Phe and Ach. However, it is still necessary to include an image showing the EC monolayer is intact and positive of CD31, as it is very rare that confluent EC monolayer can last that long in previous reports.

RESPONSE: As the reviewer suggested, we stained CD31 to the TEBVs after 1,2, and 4 weeks perfusion. The results shown that the CD31 staining is very good for at least two weeks. However, by 4 weeks of perfusion, the CD31 staining is discontinuous (Fig. S4). But the vWF staining revealed that the ECs are still survival in these 4 week vessels (Fig.S4).

7. Figure 3g: Images are too small to read. Suggest to move some to SI. From the brightfield image, it seemed that the vessel diameter and wall thickness (fibroblast-dependent) varied significantly between different conditions. These two factors can affect vessel permeability too. For permeability results to be valid, comparable lumen size and vessel wall thickness should be chosen for different conditions. Additionally, one of the highlights of the TEBV is the perfusion culture, but the author failed to show any improvement on cell phenotype of perfusion culture over static culture and perfused condition. It will be good if vessel wall permeability of static culture can also be included in this dataset.

RESPONSE: Thank you for the point. The variation in vessel dimensions was considered in the analysis of permeability since the initial fluorescence image allowed us to determine the inner diameter. Some of the error was due to variation in vessel dimensions.

To better view the results on a larger scale, we move the fluorescence image to Supplementary Results (Fig. S7) and show the permeability values in Figure 4.

As the reviewer suggested, we examined the EC orientation and shape in TEBVs under static culture and after perfusion. The results shown the ECs under static culture were round and disordered in direction. For the perfused TEBVs, ECs became longer and aligned with flow. We added these results (Fig. 2 and Fig. S4).

8. Figure 3g, 3v: As described in Methods section, the permeability of the EC barrier is calculated using the equation derived from Fick's first law, which is first described by Huxley et al. 1987. (Am J Physiol.) In the derivation of the equation, $(dI/dt)_0$ is the initial increase in fluorescent intensity as solute begins to diffuse into the field. For the equation to be valid, the fluorescent intensity change should be captured immediately after the introduction of solute into the vessel, which is usually done within 3 min as reported by other groups. In this study, the time points used was 10min and 20min,

which can't be considered as initial increase. The author should capture the changes within 3 min, quantify $(dI/dt)_0$ by fitting a linear trendline over several timepoints, and use the value of gradient for $(dI/dt)_0$ rather than obtaining multiple values for different timepoints.

RESPONSE: Thank you for the suggestion. We chose these timepoints because some previous papers reported the similar study (Max M. Gong et al, Biomaterials, 2019 (ref 51) and L. Andrique et al, Science Advances, 2019). They tested the permeability of their vessels at 15min and 10min. As suggested, we modified the analysis and added a shorter timepoint of 5min as well (Fig. 4). The overall effect was to lower the permeability, particularly for the 500 kDa dextran without cells and under static conditions.

9. Figure 4c: There are limited results on the ECM re-modeling section. ECM secretion is characterized using qRT-PCR, but there is no indication of any internal control/housekeeping gene (e.g. GAPDH) for all datasets. For qRT-PCR results to be valid, at least one internal control gene should be included as the baseline to normalize the CT value of each sample. Additionally, ECM remodeling is hallmark of atherosclerosis progression, although it was shown ECM mRNA was upregulated after 8 days recovery, actual secretion of these ECM components could be validated by immunostaining.

RESPONSE: The housekeeping gene used in this work was U6. We listed it in the Supplementary Table 1 but forgot to indicate that in the text. We added a sentence to highlight this in the method part of manuscript. We agree that matrix remodeling is important, but that would require a separate and more extensive investigation.

10. Figure 6a, 6b. Conditions with eLDL only/TNF α only should also be included for both datasets to prove whether LS blocked the drop in vasoactivity induced by eLDL or TNF α

RESPONSE: Thanks for the point. The drop in vasoactivity induced by eLDL or TNF α as well as the vasoactivity of control without treatment was shown in Fig.3c. To avoid the same results appearing twice in the figures, we didn't show them here. In the manuscript, we indicated these results should be compared to Fig.3c.

11. IL-1beta and IL-1 were used interchangeably in the manuscript, the author should make these consistent.

RESPONSE: Thanks for the point. We checked and revised the manuscript to list only IL-1 β .

Reviewer #3, an expert in in vitro blood vessel models (Remarks to the Author):

They created in vitro model of early stage atherosclerosis by fabricating, endothelizing and perfusing arteriole-scale tissue-engineered blood vessels (TEBVs).

The tube which is made by human neonatal dermal fibroblasts (hNDF) and collagen gel and endothelized in inner side using primary human endothelial colony forming cells (ECFCs). This TEBVs is useful model and have advantage (overcoming the lack of appropriate methods of the results to human cases and the expensive & time consuming) to examine the effect of genetic variants and pathogenesis of human vascular diseases.

RESPONSE: Thank you very much for the positive comments.

However, there are some concerns about the cellular component for TEBVs and interpretation on the results.

(1) They demonstrated the endothelial multilayered TEBVs replicate arteriole-mimic function, e.g. vasoreaction to phenylephrine and endothelial-dependent vasodilation. According to this evidence, hNDF

acts as vascular smooth muscle cells (VSMCs). It is better to reveal the characterization of hNDF, i.e. expression of key molecules to mediate constructive and relaxation response through α -adrenergic receptor, NO-GC respectively. Even if so, VSMCs are more ideal cell components in TEBVs. Why did they utilize fibroblasts instead of VSMCs in TEBVs?

RESPONSE: We first used fibroblasts because other reports in the literature had indicated poor contractility with primary smooth muscle cells (ref. 57 and 58). In other published experiments performed with TEBVs made with fibroblasts, we recently showed that nitric oxide donor sodium nitroprusside can elicit fibroblast relaxation (Salmon et al. 2020. *Cells* 9(5) 1292. In this revised manuscript, we successfully fabricated the TEBVs with 3 layers: endothelial layer, smooth muscle layer, and fibroblast layer. We also compared the vasoactivity function of these 3-layer TEBVs to 2-layer ones (Fig. 1g, Fig. 2g and Fig. S6).

(2) They also observed that adherent and accumulation of inflammatory cells and transformation of macrophages from circulating monocytes within the walls of TEBVs. According to their results, as compared to 2D-culture system, the phenomenon is dependent on the reaction between inflammatory cells and endothelium (EC). Please explain or demonstrate the advantage to use multilayered TEBVs instead of the EC-made perfusion tubes.

RESPONSE: While monocyte adhesion can be studied in 2D, we want to demonstrate long-term monocyte accumulation in the vessel wall as well foam cell formation by both monocytes and fibroblasts which accumulate the enzyme modified LDL. 2D studies are not designed for such subendothelial accumulation. We also examined recovery of vessel wall functions after removal of modified LDL, something not done in 2D. Overall, as noted in the Introduction, compared to the other microfluidic strategies that more focused on the endothelial cell behavior, the TEBVs disease model allows us to examine functional changes of the vessel during conditions that promote atherosclerosis including vasoactivity, burst pressure (mechanical strength), remodeling of the vascular wall (ECM), and the migration of monocytes. The obtaining of his comprehensive results benefited from the physiological structure of TEBV with multi-layers and multi-cell types, which allow the occurrence of interactions between cell-cell and cell-ECM.

(3) Mechanical factors such as shear stress are crucial for the pathogenesis of atherosclerosis. In this point, the perfusion system of TEBVs has advantage. However, the tubular size (less than 1 mm) of TEBVs is far different compared to human atherosclerotic vessels such as coronary artery (2-5 mm in diameter). Please explain the substitutional points and the scale-based limitation between TEBVs and actual human vessels.

RESPONSE: Thank you for the point. Yes, we agree that the TEBVs used here are smaller than real coronary artery. We designed the TEBVs with this dimension (OD: 1 mm, ID: 0.6 mm) because it is at the same level of a small artery and larger arteriole (mm level) and the perfusion chamber are small enough to be imaged clearly under stereomicroscope with 4 vessels. Accordingly, we reduced the flowrate depended on the TEBV dimension to mimic similar shear stress to real vessel. Furthermore, if needed, the dimension of the TEBVs could be changed flexibly. We only need to extend the dimension of the mold and use the mandrels with bigger diameter. The flexibility of fabrication method is one of the advantages of our work too.

As now noted in the Discussion, the human TEBV microphysiological system can isolate specific effects that are difficult to assess *in vivo*, such separate a direct effect of statins on the vessel wall from their cholesterol lowering effect. By keeping the diameter small, media volumes and flow rates can be maintained at small values to replicate physiological shear stresses (ml/min or less). Small volumes are

ideal to measure any metabolites, secreted molecule or cytokines produced and when evaluating drug responses. A limitation to any microphysiological system, either TEBV or microfluidic device, to simulate *in vivo* conditions for larger vessels is that dynamic similarity results in higher shear stresses than those occurring in human blood vessels, which may limit leukocyte adhesion and endothelial function. If, however, the focus is upon the shear stress acting on endothelium, then the Reynolds numbers and Womersley will be less than those *in vivo*, potentially leading to greater leukocyte adhesion and transmigration.

Minor comments

(1) In figure 1, it is hard to distinguish the colors indicating fibroblast and eLDL-activated fibroblasts. It is better to consider the color design for being kind to color-blind readers

RESPONSE: We have adjusted colors so it should be easier to identify eLDL-activated cells.

(2) In figure 3 (g, h), what does it compare to? Please indicate the number of explanation.

RESPONSE: Figures 3g, h are now Figures S4 a,b. We clearly indicate that panel a refers to FITC-labeled 500 kD dextran and panel b is for FITC labeled goat IgG. Each of the panels i through v is denoted in the legend.

(3) In figure 4 legend, TNF α may be fixed to TNF α (alpha).

RESPONSE: Done.

(4) In figure 5 (k, l), hard to follow up the results. In figure 5k, solely green-signals are observed. CD81+ monocytes (green) are detected within total red-labeled monocytes? MFI of double positive cells are measured?

RESPONSE: Figures 5 j-l are now Figures 7 a-d. We revised the figure legends and associated text to clarify the experimental conditions and results. All of the CD80+ monocytes (green signal) were positive for cell tracker red.

In addition, we checked the entire manuscript again and revised all the typographical errors.

REVIEWERS' COMMENTS

Reviewer #1 (Remarks to the Author):

The authors invested major efforts to address my previous comments. The changes performed and additional experiments addressed most of my concerns successfully.

Reviewer #3 (Remarks to the Author):

The article is revised appropriately in response to reviewers suggestions and questions.

The revised article has been improved with additional valuable data.

Just minor suggestions;

In Figure 2 and results, it is better to use hSMC instead of SMC , comparing with hNDF.

Please add the explanation of the abbreviation,i.e. SMC in figure 2 legend.

Reviewer #1 (Remarks to the Author):

The authors invested major efforts to address my previous comments. The changes performed and additional experiments addressed most of my concerns successfully.

Reply: Thank you for the comments.

Reviewer #3 (Remarks to the Author):

The article is revised appropriately in response to reviewers suggestions and questions. The revised article has been improved with additional valuable data.

Reply: Thank you for the comments.

Just minor suggestions;

In Figure 2 and results, it is better to use hSMC instead of SMC, comparing with hNDF. Please add the explanation of the abbreviation,i.e. SMC in figure 2 legend.

Reply: We changed the term SMC to hSMC in Figure 2, Results part and Supplementary Material. Further, we added the explanation of the abbreviation in Figure 2 legend as “hNDF: human neonatal dermal fibroblasts; hSMC: human smooth muscle cells”.